# EZH2 inhibition decreases neuroblastoma proliferation and *in vivo* tumor growth

**Laura V. Bownes**[1�], **Adele P. Williams**[1�], **Raoud Marayati**[1], **Laura L. Stafman**[1], **Hooper Markert**[1], **Colin H. Quinn**[1], **Nikita Wadhwani**[1], **Jamie M. Aye**[2], **Jerry E. Stewart**[1], **Karina J. Yoon**[3], **Elizabeth Mroczek-Musulman**[4], **Elizabeth A. Beierle**[1]*

**1** Division of Pediatric Surgery, Department of Surgery, University of Alabama at Birmingham, Birmingham, Alabama, United States of America, **2** Division of Hematology and Oncology, Department of Pediatrics, University of Alabama at Birmingham, Birmingham, Alabama, United States of America, **3** Division of Pharmacology and Toxicology, University of Alabama at Birmingham, Birmingham, Alabama, United States of America, **4** Department of Pathology, Children's of Alabama, Birmingham, Alabama, United States of America

l These authors contributed equally to this work.

\* elizabeth.beierle@childrensal.org

## Abstract

Investigation of the mechanisms responsible for aggressive neuroblastoma and its poor prognosis is critical to identify novel therapeutic targets and improve survival. Enhancer of Zeste Homolog 2 (EZH2) is known to play a key role in supporting the malignant phenotype in several cancer types and knockdown of EZH2 has been shown to decrease tumorigenesis in neuroblastoma cells. We hypothesized that the EZH2 inhibitor, GSK343, would affect cell proliferation and viability in human neuroblastoma. We utilized four long-term passage neuroblastoma cell lines and two patient-derived xenolines (PDX) to investigate the effects of the EZH2 inhibitor, GSK343, on viability, motility, stemness and *in vivo* tumor growth. Immunoblotting confirmed target knockdown. Treatment with GSK343 led to significantly decreased neuroblastoma cell viability, migration and invasion, and stemness. GSK343 treatment of mice bearing SK-N-BE(2) neuroblastoma tumors resulted in a significant decrease in tumor growth compared to vehicle-treated animals. GSK343 decreased viability, and motility in long-term passage neuroblastoma cell lines and decreased stemness in neuroblastoma PDX cells. These data demonstrate that further investigation into the mechanisms responsible for the anti-tumor effects seen with EZH2 inhibitors in neuroblastoma cells is warranted.

## Introduction

Neuroblastoma, a neural crest tumor, continues to be responsible for over 15% of all pediatric cancer deaths [1]. Children with high-risk disease fare the most poorly, and minimal advances have been made in improving their outcomes [2]. Novel pathways and targets must be investigated to provide innovative therapeutic interventions for these children.

Enhancer of Zeste Homolog 2 (EZH2), a SET domain-containing histone methyltransferase, has become a target of interest in numerous malignancies including glioblastoma,

**Data Availability Statement:** All relevant data are within the manuscript and its Supporting information files.

**Funding:** This work was partially funded by institutional grants from the National Cancer Institute including T32 CA229102 (https://grantome.com/grant/NIH/T32-CA229102-01)(LV Bownes, R Marayati), T32 CA183926 (https://grantome.com/grant/NIH/T32-CA183926-01)(AP Williams), T32 CA091078 (https://grantome.com/grant/NIH/T32-CA091078-14) (LL Stafman), 5T32BM00836 (CH Quinn), and P30 AR048311 and P30 AI27667 to the University of Alabama at Birmingham, Flow Cytometry Core (https://www.uab.edu/medicine/cfar/core-facilities/basic-research-core/flow-cytometry-core). Funding was also provided by the Lombardi Cancer Research Fund/Starr Children's Fund (https://www.lombardifoundation.org/home-annual-report), Sid Strong Foundation (https://sidstrongfound.org/), Elaine Roberts Foundation (https://www.elainerobertsfoundation.org/), and Open Hearts Overflowing Hands (https://openhandsoverflowinghearts.org/) (EA Beierle, JM Aye). The funders had no role in study design, data collection and analysis, decision to publish, or preparation of the manuscript.

**Competing interests:** The authors have declared that no competing interests exist.

leukemia, ovary, lung, colon, and breast cancers [3]. EZH2 belongs to the catalytic subunit of the polycomb repressive complex 2 (PRC2) along with two additional proteins, embryonic ectoderm development (EED) and suppressor of zeste 12 (SUZ12). PRC2 mediates gene silencing primarily by regulating chromatin structure, and consequently, gene expression. PRC2 catalyzes trimethylation by removing a methyl group from S-adenosyl methionine (SAM) to histone H3 lysine 27 (H3K27me3), which is an important epigenetic factor determining stem cell differentiation [3,4].

The oncogenic role of EZH2 is defined by the methylation of H3K27 of tumor suppressor genes, and EZH2 has been implicated in other tumorigenic pathways including β-catenin, Ras, and nuclear factor kappa B (NF-κB) [3,5]. Overexpression of EZH2 has been shown to be a marker of advanced disease in prostate and breast cancers while inactivating mutations suggest a worse prognosis in myeloid neoplasms [3,6]. In neuroblastoma, it has been shown that high expression of EZH2 correlated with poor prognosis [7]. Other investigators demonstrated that suppression of PRC2 subunits in neuroblastoma decreased tumor growth *in vitro* and *in vivo* [8]. EZH2 upregulation has been demonstrated to activate the Src kinase pathway in cancer, in addition to various other kinase dependent pathways [3]. Due to the evidence for the role of EZH2 in tumorigenesis, several EZH2 inhibitors have been developed and are in various stages of evaluation, including GSK343, which is a SAM-competitive inhibitor of EZH2 [5].

Based on the findings supporting the role of EZH2 in promoting tumorigenesis and its correlation with poor outcome in neuroblastoma, we hypothesized that blocking EZH2 function in neuroblastoma cell lines would result in decreased proliferation and motility *in vitro* and impede tumor growth *in vivo*. Further, EZH2 has been shown to activate Src, a protein kinase upstream from focal adhesion kinase (FAK). FAK, a nonreceptor protein tyrosine kinase, has been shown to play an important role in neuroblastoma tumorigenesis [9–11]. We further hypothesized that EZH2 inhibition with GSK343 would affect FAK expression.

## Materials and methods

### Cells and cell culture

The human neuroblastoma cell lines SK-N-AS (CRL-2137) and SK-N-BE(2) (CRL-2271) were obtained from American Type Culture Collection (ATCC, Manassas, VA). The isogenic *MYCN-* SH-EP and *MYCN+* WAC(2) cell lines were a generous gift from Dr. M. Schwab (Deutsches Krebsforschungszentrum, Heidelberg, Germany), and have been described previously in detail [12]. Cell lines were maintained under standard culture conditions at 37˚C and 5% $CO_2$ in the following media: Dulbecco's modified Eagle's medium (DMEM, 30–2601, ATCC) containing 10% fetal bovine serum (FBS, Hyclone, Suwanee, GA), 4 mM L-glutamine (Thermo Fisher Scientific Inc., Waltham, MA), 1 μM non-essential amino acids, and 1 μg/mL penicillin/streptomycin (Gibco, Carlsbad, CA) for SK-N-AS cells; and a 1:1 mixture of Eagle's minimum essential medium (30–2003, ATCC) and Ham's F-12 medium (30–2004, ATCC) with 10% fetal bovine serum (Hyclone), 2 mM L-glutamine (Thermo Fisher Scientific), 1 μM non-essential amino acids, and 1 μg/mL penicillin/streptomycin (Gibco) for SK-N-BE(2) cells. SH-EP and WAC(2) cell lines were maintained in RPMI 1640 medium (Corning Inc., Corning, NY) supplemented with 10% fetal bovine serum (Hyclone) and 1 μg/mL penicillin/streptomycin (Gibco). All four cell lines were verified within the last 12 months using short tandem repeat analysis [University of Alabama at Birmingham (UAB) Genomics Core, Birmingham, AL], and deemed free of Mycoplasma infection. The human neuroblastoma patient-derived xenolines (PDXs), COA3 and COA6, were utilized. These PDXs have been previously described in detail [9]. Briefly, individual PDX cells were

obtained by dissociating the xenograft tumors using a Tumor Dissociation Kit (Miltenyi Biotec, San Diego, CA) per manufacturer's protocol, and resuspending in neurobasal medium (NB, Life Technologies, Carlsbad, CA) supplemented with B-27 without Vitamin A (Life Technologies), N2 (Life Technologies), L-glutamine (2 mM), epidermal growth factor (10 ng/mL, Miltenyi Biotec), fibroblast growth factor (10 ng/mL, Miltenyi Biotec), amphotericin B (250 μg/mL), and gentamicin (50 μg/mL). Following dissociation, cells were maintained at 37˚C with 5% $CO_2$ overnight prior to use for experimentation. Real-time PCR was performed to assess the percentage of human and mouse DNA contained in the PDXs to ensure that the tumors did not harbor excessive murine contamination and had not undergone a transformation to a murine tumor (TRENDD RNA/DNA Isolation and TaqMan QPCR/ Genotyping Core Facility, UAB, Birmingham, AL). PDX cells were also verified within the last 12 months using short tandem repeat analysis (UAB Genomics Core). Details regarding patient demographics are provided in S1 Table.

## Establishing patient-derived xenolines

The PDX program was approved by the University of Alabama at Birmingham (UAB) Institutional Review Board (X130627006), and the studies were conducted at the University of Alabama at Birmingham, Children's Hospital of Alabama. From November 2013 through January 2014, pediatric patients (0–21 years) with tumors suspected to be neuroblastoma were identified in the pediatric hematology/oncology or surgery clinics or after admission to the Children's Hospital of Alabama. Parents of the children were approached, and the study was thoroughly explained to them. Written informed consent was obtained from the patients' parents prior to collection of tumor tissue. When appropriate, written assent was also obtained from the patients. Consents and assents were witnesses by adults who were independent from the research team and not involved in the studies. There were no specific inclusion/exclusion criteria for the study. COA3 was a *MYCN* amplified, high-risk primary tumor originating in a female child and COA6 was a *MYCN* amplified, high-risk primary tumor originating in a male child (S1).

For the PDX studies, all animal experiments were approved by the University of Alabama at Birmingham (UAB) Institutional Animal Care and Use Committee (IACUC-09186) and were conducted within institutional, national and NIH guidelines. Neuroblastoma tumor tissue was obtained fresh from patients with primary tumors and kept in Roswell Park Memorial Institute (RPMI) 1640 medium on ice for transport. Tumor chunks were transplanted subcutaneously into the flank of female NOD SCID mice (Envigo, Prattville, AL). Tumor volumes were measured with calipers and calculated with the standard formula (width$^2$ × length)/2, where the width was the smallest measurement. When tumors reached 2000 mm$^3$, they were harvested, chopped, and sequentially implanted from animal to animal for xenoline expansion. Separate portions of the tumor were dissociated for experiments.

## Reagents and antibodies

GSK343 (S7164) was purchased from Selleckchem (Houston, TX). Primary antibodies used for Western blotting included the following: rabbit anti-EZH2 (5246S), anti-H3 (4499S, clone D1H2), anti-FAK (71433S) and anti-H3K27me3 (9733S, clone C36B11) from Cell Signaling (Danvers, MA); anti-FAK(C-20) (sc-558) from Santa Cruz (Santa Cruz Biotechnology, Dallas, TX); anti-FAK (05–537, clone 4.47) from Millipore (EMD Millipore, Burlington, MA); and mouse anti-β-actin from Sigma (A1978, Sigma Aldrich, St. Louis, MO), anti-GAPDH (MAB374, clone 6C5) from Millipore and anti-MYCN from Santa Cruz (sc-53993).

## Immunoblotting

Following treatment, cells were lysed on ice in a buffer consisting of 50 mM Tris-HCl (pH 7.4), 150 mM NaCl, 1 mM EDTA, 1% Triton x-100, 1% sodium deoxycholate, 0.1% SDS, phosphatase inhibitor (P5726, Sigma Aldrich), protease inhibitor (P8340, Sigma Aldrich), and phenylmethylsulfonyl fluoride (PMSF, P7626, Sigma Aldrich) for 30 minutes. Lysates were centrifuged at 14 000 rpm for 30 minutes at 4°C. Protein concentrations were determined using a Micro BCA™ Protein Assay Kit (Thermo Fisher Scientific). Proteins were separated on SDS-PAGE gels by electrophoresis and transferred to Immobilon®-P polyvinylidene fluoride (PVDF) transfer membrane (EMD Millipore). Precision Plus Protein Kaleidoscope Standards (161–0375, Bio-Rad, Hercules, CA) molecular weight markers were used to confirm expected size of target proteins. Antibodies were used per the manufacturers' recommended protocol. Samples were visualized by enhanced chemiluminescence (ECL) using Luminata Classico or Luminata Crescendo Western horseradish peroxidase (HPR) substrates (EMD Millipore). Anti-β-actin or vinculin served as an internal control to ensure equal protein loading.

## Cell proliferation and viability assays

Proliferation was assessed using the CellTiter 96® Aqueous One Solution Cell Proliferation assay (Promega, Madison, WI). Cells were treated with increasing concentrations of GSK343 (0, 5, 15, 25 μM) for 24 hours and plated ($5 \times 10^3$ cells/well) onto 96-well plates. After 24 hours, CellTiter 96® dye (10 μL) was added to each well and the absorbance was measured at 490 nm using a microplate reader (Epoch Microplate Spectrophotometer, BioTek Instruments, Winooski, VT). Viability was evaluated using an alamarBlue® assay (Thermo Fisher Scientific). Cells were treated with increasing concentrations of GSK343 (0, 5, 15, 25 μM) for 24 hours, plated ($1.5 \times 10^3$ cells) onto 96-well plates and after 24 hours, 10 μL of alamarBlue® dye was added to each well. The plates were read at using a microplate reader (Epoch Microplate Spectrophotometer) to detect the absorbance at 570 nm, using 600 nm as a reference wavelength. Proliferation and viability experiments were completed with at least three biologic replicates and data reported as fold change ± standard error of the mean (SEM).

## Monolayer wound healing (scratch) assay

Effects of GSK343 on SK-N-AS and SK-N-BE(2) cell migration was evaluated utilizing a monolayer wound healing assay. Cells were treated for 24 hours with GSK343 (0, 15 μM) then plated and allowed to reach 80% confluence. A sterile 200 μL pipette tip was employed to make a standard scratch in the well. Images were obtained of the scratch wound at 0, 12, and 24 hours. The area of the open wound in pixels was quantified using the ImageJ MRI Wound Healing Tool [13]. Data were reported as fold change scratch area ± SEM and compared between groups. Monolayer wounding assays were not completed on COA6 cells as they do not attach in cell culture.

## Animal statement

Animal experiments were approved by the University of Alabama at Birmingham (UAB) Institutional Animal Care and Use Committee (IACUC-09355) and were conducted within institutional, national, and NIH guidelines. The Department of Comparative Medicine through the Animal Resources Program (ARP) at UAB manages a fully accredited (AAALAC) animal laboratory. 6-week-old female athymic nude mice (Envigo, Prattville, AL) were maintained in a pathogen-free facility with 12-hour light/dark cycles, static conventional housing, and *ad libitum* access to Harlan Rodent Diet® Teklad 4% fat mouse/rat chow (Envigo) and water. Mice

were provided cardboard tubes and wood chew sticks for environmental enrichment. Mice were humanely euthanized by a two-step method by $CO_2$ inhalation followed by cervical dislocation. The *in vivo* tumors were measured three times per week, and weights measured weekly. After tumor cell injections, the animals were monitored on a daily basis by both our laboratory staff and the animal welfare veterinary staff. Animals that were humanely euthanized with the method listed above were done so for the following IACUC criteria: (i) body condition score less than 2; (ii) weight loss greater than 10% of their body weight; (iii) loss of grooming behavior; (iv) cessation of eating or drinking; (v) tumor size over 2 $cm^3$; (vi) ulcerated tumor; (vii) tumor hindering ambulation. The animals did not undergo any surgical intervention prior to euthanasia for tumor harvest. We utilized 14 animals for the *in vivo* tumor growth study outlined below and 21 animals for propagating the xenolines.

### *In vivo* tumor growth

SK-N-BE(2) ($1.8 \times 10^6$) cells in 25% Matrigel™ (Corning Inc.) were injected into the right flank of 6-week-old, female, athymic nude mice (n = 7 per group) (Envigo). Tumors were measured twice weekly and tumor volumes calculated with the formula (width$^2 \times$ length)/2, with width being the smallest measurement. Once tumors were palpable (100 $mm^3$), the animals were randomized to receive either 100 μL of sterile phosphate buffered saline (PBS, vehicle) or GSK343 (10 mg/kg/day in 100 μL PBS) once daily via intraperitoneal (IP) injection for 21 days. The GSK343 dosing was based on previously published animal data [4,14]. Flank tumors were measured three times per week, and tumor volumes calculated as described above. The animals were humanely euthanized after 21 days of treatment or when IACUC parameters were met.

### Cell migration and invasion assays

Cell migration and invasion assays were performed with COA3 and COA6 human neuroblastoma PDX cells using 6.5 mm Transwell® inserts with 8 μM pore polycarbonate membrane (Corning Inc.). For both assays, the bottoms of the inserts were coated with collagen Type I (10 mg/mL, 50 μL) for 4 hours at 37˚C. Additionally, for invasion assays, the top sides of the inserts were coated with Matrigel™ (1 mg/mL, 50 μL, Corning Inc.) for 4 hours at 37˚C. Cells were pretreated with GSK343 (0, 10 μM) for 24 hours and then $4.0 \times 10^4$ cells plated into the top portion of the insert. The bottom well contained 10% FBS as a chemo-attractant. Cells were allowed to migrate for 72 hours and invade for 1 week. The inserts were then fixed in 3% paraformaldehyde and stained with crystal violet. SPOT Basic 5.2 (Diagnostic Instruments Inc., Sterling Heights, MI) imaging software was used to photograph the inserts at predetermined locations with a microscope at 40 × and cells quantified using ImageJ software (Ver 1.49, available online at http://imagej.nih.gov/ij). Experiments were repeated in triplicate and migration and invasion reported as fold change ± SEM.

### *In vitro* limiting dilution tumorsphere assay

To determine if GSK343 disrupted the stem cell-like phenotype, tumorsphere forming ability was assessed with *in vitro* limiting dilution assays [15]. Conditioned COA3 and COA6 media was harvested from untreated cells in culture. Single cells were plated in conditioned media onto 96-well low attachment plates using serial dilutions with 5000, 1000, 100, 50, 20, 10, and 1 cell per well. Cells were then treated with GSK343 (0, 5 μM). A week later, the number of wells containing tumorspheres in each row was counted by a single researcher and the results were analyzed using the extreme limiting dilution analysis software (http://bioinf.wehi.edu.au/software/elda/).

## Real-time PCR (qPCR)

The effects of GSK343 on mRNA abundance of known stemness markers were assessed using qPCR. COA6 cells were treated with GSK343 (0, 5 μM) for 72 hours and total cellular RNA was extracted using the RNeasy kit (Qiagen) according to the manufacturer's protocol. For synthesis of cDNA, 1 μg of RNA was used in a 20-μl reaction mixture utilizing an iScript cDNA Synthesis kit (Bio-Rad) according to the supplier's instructions. Resulting reverse transcription products were stored at -20˚C until further use. For qPCR, SsoAdvanced™ SYBR® Green Supermix (Bio-Rad) was utilized according to manufacturer's protocol. Probes specific for the stemness markers Oct4, Nanog, and Sox2, as well as for actin B were obtained (Applied Biosystems, Foster City, CA). qPCR was performed with 10 ng cDNA in 20 μL reaction volume. Amplification was done using an Applied Biosystems 7900HT cycler (Applied Biosystems). Cycling conditions were 95˚C for 2 min, followed by 39-cycle amplification at 95˚C for 5 secs and 60˚C for 30 sec. Experiments were repeated with at least three biologic replicates, and samples were analyzed in triplicate with actin B utilized as an internal control. Data were calculated utilizing the ΔΔCt method [16] and are reported as mean fold change ± SEM.

## Immunofluorescence

Cells were plated on glass chamber slides and allowed to attach for 24 hours, treated for 24 hours, then fixed with 4% paraformaldehyde. Cells were permeablized with 0.15% Triton X-100, and the first primary antibody (anti-EZH2, 5246S, Cell Signaling) was added and incubated at room temperature (RT) for 1 hour followed by the addition of the second primary antibody (anti-FAK 4.47, EMD Millipore) that was also incubated for 1 hour at RT. The Alexa Fluor 488 secondary antibody (goat anti-rabbit, A-11034, Invitrogen) was added for 45 minutes at RT. After washing, the second secondary antibody, Alexa Fluor 594 (goat anti-mouse, A-11005, Invitrogen), was added and incubated as above. Prolong® Gold antifade reagent with DAPI (P36931, Invitrogen) was used for mounting. Imaging was performed with a Zeiss LSM 710 Confocal Scanning Microscope with Zen 2008 software (Carl Zeiss MicroImaging, LLC, Thornwood, NY) using a 63× objective with a zoom of 0.9. Fifteen cells per sample were analyzed with 20 images each. Manders overlap coefficients were calculated [17]. Manders coefficients have a value between 0 and 1, with 0 = no overlap and 1 = perfect overlap. These coefficients provide the proportion of overlap of each channel with the other.

## Immunoprecipitation

Whole cell lysates were subjected to immunoprecipitation followed by immunoblotting. Briefly, cell lysates were added to Protein A/G agarose beads (Santa Cruz Biotechnology), incubated at 4˚C for 1 hour and then spun in the cold to pre-clear the cell lysate. The primary antibody was added to the supernatant and incubated overnight at 4˚C. Protein A/G agarose beads were added and incubated for 3 hours and washed on ice. The samples were heated to 100˚C for 5 minutes and centrifuged for 1 minute at 14 000 g. Samples (500 μg protein) were loaded onto SDS-PAGE gels and electrophoresed as described above. Rabbit IgG was included on the immunoblots as a control.

## Statistical analyses

*In vitro* experiments were performed at a minimum of three biologic replicates. Data reported as the mean ± SEM. Parametric data between groups was compared using an analysis of variance (ANOVA) or Student's t-test as appropriate. Non-parametric data were

analyzed with Mann-Whitney U test (Wilcoxon Rank Sum Test). Statistical significance was defined as p $\leq$ 0.05.

## Results

### EZH2 inhibitor, GSK343, decreased tri-methylation of Histone 3 at Lysine 27

Four long-term passage neuroblastoma cell lines, SK-N-AS, SK-N-BE(2), SH-EP and WAC(2) and two human neuroblastoma PDXs, COA 3 and COA6, were utilized for experimentation. Documentation of expression of EZH2 and the downstream target, tri-methylation of Histone 3 at Lysine 27 (H3K27me3) was necessary prior to further investigation. Immunoblotting revealed that EZH2 and H3K27me3 were both present in both the long-term passage neuro-blastoma cell lines (Fig 1A) and the human neuroblastoma PDX cells (Fig 1B and 1C). Treatment with increasing concentrations of GSK343 for 24 hours decreased the tri-methylation of Histone 3 at Lysine 27 and the expression of EZH2 in the PDXs (Fig 1B and 1C), but did not change H3 expression (Fig 1A–1C).

### GSK343 decreased neuroblastoma proliferation, viability, and motility

We proceeded to examine the effects of GSK343 on the phenotype of neuroblastoma cells. Viability and proliferation were evaluated with alamarBlue® and CellTiter96®, respectively. Inhibition of EZH2 with GSK343 affected viability in SK-N-AS, SK-N-BE(2) and SH-EP neu-roblastoma cell lines (Fig 2A). GSK343 significantly decreased neuroblastoma proliferation (Fig 2B). Information on median lethal dose provided in supplemental S2 Table.

EZH2 has been shown to affect cell migration and invasion in other malignancies [18–20], leading us to examine the role of EZH2 inhibition in neuroblastoma cell motility. Cell motility was evaluated with a monolayer wound healing (scratch assay) in SK-N-AS (Fig 2C), SK-N-BE (2) (Fig 2D and 2G), SH-EP (Fig 2E) and WAC(2) (Fig 2F). There was a significant decrease in the area of the scratch closed, indicating decreased cell motility in cells treated with GSK343. These findings showed that EZH2 affected neuroblastoma viability, proliferation, and motility.

### GSK343 decreased tumor growth *in vivo*

To demonstrate that GSK343 inhibition of EZH2 was relevant *in vivo*, we proceeded to an ani-mal model. SK-N-BE(2) cells were injected into the right flank of athymic nude mice and ani-mals were monitored for tumor growth. Tumor volumes were measured three times per week. Fold change in tumor volume was significantly smaller in the GSK343 treated animals when compared vehicle treated controls (Fig 3A). Further, relative tumor growth was significantly decreased in mice treated with GSK343 compared to those treated with vehicle (Fig 3B). The

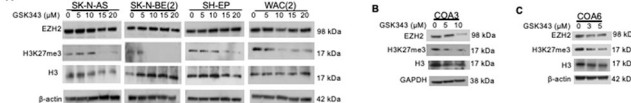

**Fig 1. EZH2 inhibitor, GSK343, decreased tri-methylation of Histone 3 at Lysine 27.** (A) Immunoblotting revealed EZH2 and H3K27me3 expression in long-term passage neuroblastoma cell lines. Treatment with increasing concentrations of GSK343 for 24 hours led to a decrease in EZH2 expression with increasing doses. Increasing doses of GSK343 decreased tri-methylation of the EZH2 downstream target, Histone 3 at Lysine 27 (H3K27me3). H3 expression was unchanged. (B, C) Similar to long-term passage cell lines, immunoblotting of the human neuroblastoma PDXs demonstrated a decrease in EZH2 expression with the inhibitor GSK343 as well as a decrease in tri-methylation of Histone 3 at Lysine 27, but H3 was not significantly changed.

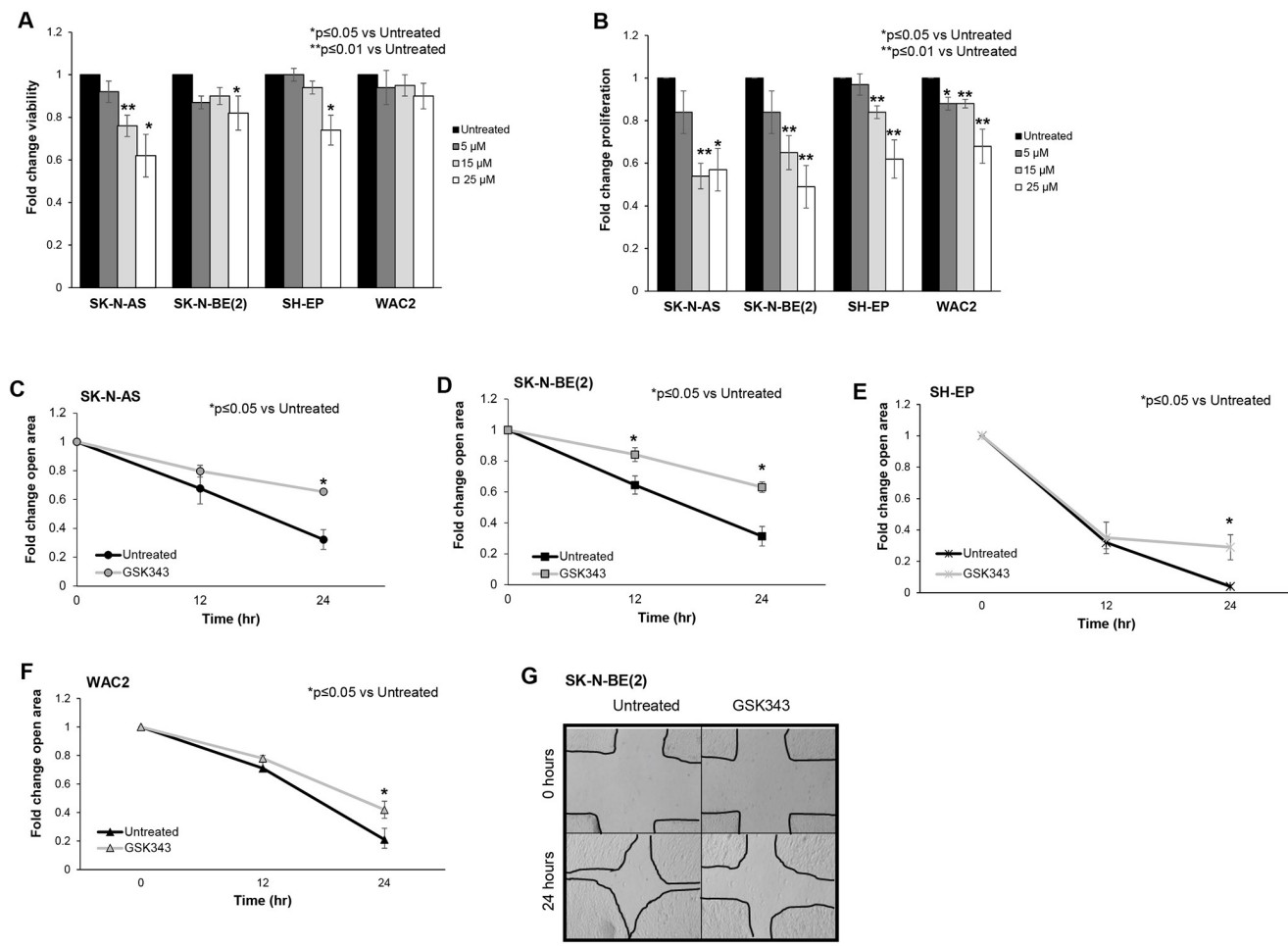

**Fig 2. GSK343 decreased neuroblastoma viability, proliferation, and motility.** (A) SK-N-AS, SK-N-BE(2), SH-EP and WAC(2) cells ($1.5 \times 10^3$ cells) were treated with increasing concentrations of GSK343 (0, 5, 15, 25 μM) for 24 hours and viability was measured using alamarBlue® assay. GSK343 treatment resulted in decreased viability in SK-N-AS, SK-N-BE(2), and WAC(2) cells. (B) SK-N-AS, SK-N-BE(2), SH-EP and WAC(2) cells ($1.5 \times 10^3$ cells) were treated with increasing concentrations of GSK343 (0, 5, 15, 25 μM) for 24 hours and proliferation was measured with using CellTiter® assay. Proliferation was significantly decreased following treatment with GSK343 in all cell lines. (C) SK-N-AS, (D) SK-N-BE(2), (E) SH-EP and (F) WAC(2) cells were treated for 24 hours with GSK343 (15 μM) then plated and allowed to reach 80% confluence. A standard scratch was made in each well and images of the scratch were obtained at 0, 12, and 24 hours. The area of the open wound in pixels was quantified using the ImageJ MRI Wound Healing Tool. By 24 hours, there was a significant decrease in the area of the scratch healed (indicating decreased motility) in cells treated with GSK343 compared to untreated cells. Data were reported as fold change scratch area ± SEM and compared between groups. (G) Representative photomicrographs of SK-N-BE(2) cell wounding assays. GSK343 treated cells (*right panel*) demonstrated significant reduction in ability to heal the scratch compared to untreated cells (*left panel*). Data represent at least three biologic replicates.

GSK343 treatment was well tolerated as demonstrated by constant weight gain in the treated animals (Fig 3C).

## GSK343 decreased neuroblastoma PDX viability, proliferation, and motility

Since long-term passage cell lines may not completely reflect the actual clinical condition [21], we employed human neuroblastoma PDX cells to further study the effects of EZH2 inhibition on neuroblastoma. We again examined viability and proliferation. Viability (Fig 4A) and proliferation (Fig 4B) were significantly decreased following treatment with

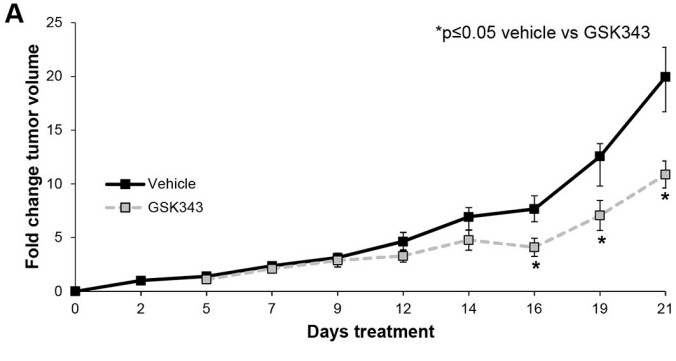

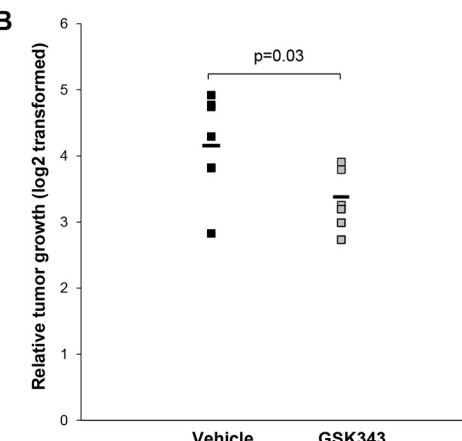

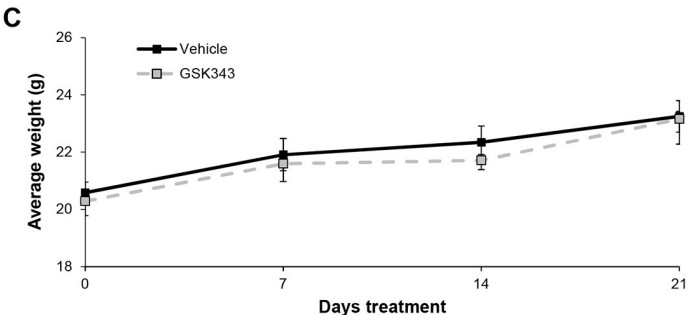

**Fig 3. GSK343 treatment decreased *in vivo* tumor volume and growth.** SK-N-BE(2) ($1.8 \times 10^6$) cells were injected subcutaneously into the right flank of 6-week-old, female, athymic nude mice. Once tumors were palpable (100 mm³), mice were randomized to receive either 100 μL of sterile PBS (vehicle, n = 7) or GSK343 (10 mg/kg/day in 100 μL PBS, n = 7) once daily via intraperitoneal (IP) injection for 21 days. Tumor volumes were monitored three times per week. (A) Animals treated with GSK343 had significantly decreased fold change in tumor volumes compared to mice treated with vehicle. (B) Animals treated with GSK had significantly decreased relative tumor growth compared to those treated with vehicle. (C) Mice were weighed at the beginning of the experiment, weekly, and at the time of euthanasia. There was no significant difference in weights between animals that received vehicle compared to those that received GSK343.

GSK343. Information on median lethal dose provided in S2. To investigate whether GSK343 affected cell motility, modified Boyden chamber assays were utilized to examine migration and invasion in COA 3 and COA6 PDX cells. There was a significant decrease in cell migration (Fig 4C) and invasion (Fig 4D) with GSK343 treatment. Representative photomicrographs of migration and invasion inserts for COA6 cells are presented in Fig 4E. Black scale bars represent 100 μm.

## GSK343 decreased stemness in neuroblastoma PDX cells

EZH2 has been shown to play a role in maintaining the cancer stem cell-like phenotype in other cancer types [3]. We wished to determine if GSK343-induced EZH2 inhibition reduced the stem cell-like phenotype in the neuroblastoma PDXs, COA3 and COA6. An extreme limiting dilution assay was used to investigate the effect of GSK343 on the ability of PDX cells to form tumorspheres, a marker of cancer cell stemness. When compared to untreated cells, PDX cells treated with GSK343 (0, 5 μM) had significantly decreased sphere forming ability (Fig 5A and 5B), indicating that the cells were less stem cell-like. The mRNA abundance of the stem cell makers, Oct4, Nanog, and Sox2 was examined with qPCR in the COA6 cells treated with GSK343 (0, 5 μM). At 72 hours, GSK343 resulted in a significant decrease in the mRNA abundance of all three markers (Fig 5C). These data provided evidence that GSK343 decreased the cancer stem cell-like phenotype in these human neuroblastoma PDX cells.

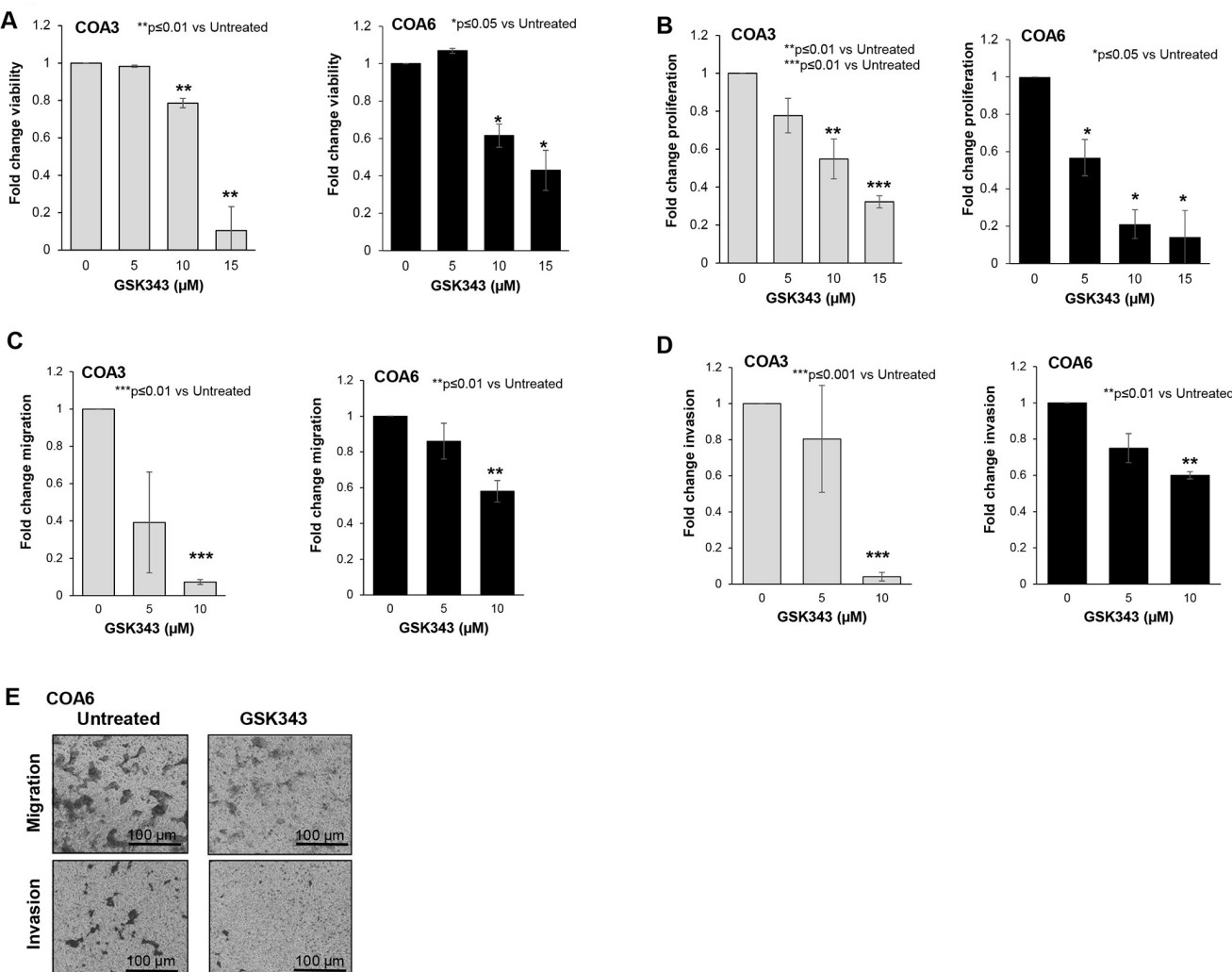

**Fig 4. GSK343 decreased viability and proliferation in human neuroblastoma PDX cells.** (A) COA3 and COA6 ($1.5 \times 10^3$) cells were treated with increasing concentrations of GSK343 (0, 5, 15, 25 μM) for 24 hours and viability was measured with alamarBlue® assay. GSK343 treatment resulted in significantly decreased viability. (B) COA3 and COA6 ($1.5 \times 10^3$) cells were treated with increasing concentrations of GSK343 (0, 5, 15, 25 μM) for 24 hours and proliferation was measured using CellTiter® assay. Proliferation was significantly decreased following treatment with GSK343.(C) COA3 and COA6 cells were treated for 24 hours with increasing doses of GSK343 (0, 5, 10 μM) and plated ($4.0 \times 10^4$ cells) in Transwell® inserts. Cells were allowed to migrate for 72 hours. Treatment with GSK343 resulted in significantly decreased migration. (D) After 24 hours of treatment with increasing concentrations of GSK343 (0, 5, 10 μM), COA3 and COA6 cells ($4.0 \times 10^4$) were plated in Transwell® inserts and allowed to invade for 1 week through a Matrigel™ layer. Invasion was significantly decreased in the cells treated with GSK343. (E) Representative photomicrographs of migration and invasion inserts for untreated (*left panel*) and GSK343 treated (*right panel*) COA6 cells. Scale bars represent 100 μm. Migration was reported as mean percent area positive ± SEM and invasion reported as mean number of cells ± SEM. Experiments were repeated with at least three biologic replicates.

## Inhibition of EZH2 led to decreased FAK expression

EZH2 has been shown to be associated with Src kinase [3,22,23]. Src activation resulted in increased EZH2 expression, and Src inhibition decreased EZH2 expression in mammary malignancies [3]. We hypothesized that FAK, a Src downstream kinase, may also be affected by GKS343. FAK has been shown to correlate with aggressive neuroblastoma [24], and FAK inhibition resulted in decreased neuroblastoma tumor growth *in vivo* [25]. Immunoblotting revealed a decrease in FAK expression with increasing doses of GSK343 in long-term passage neuroblastoma cell lines (Fig 6A). Based on these results, dual immunofluorescence staining

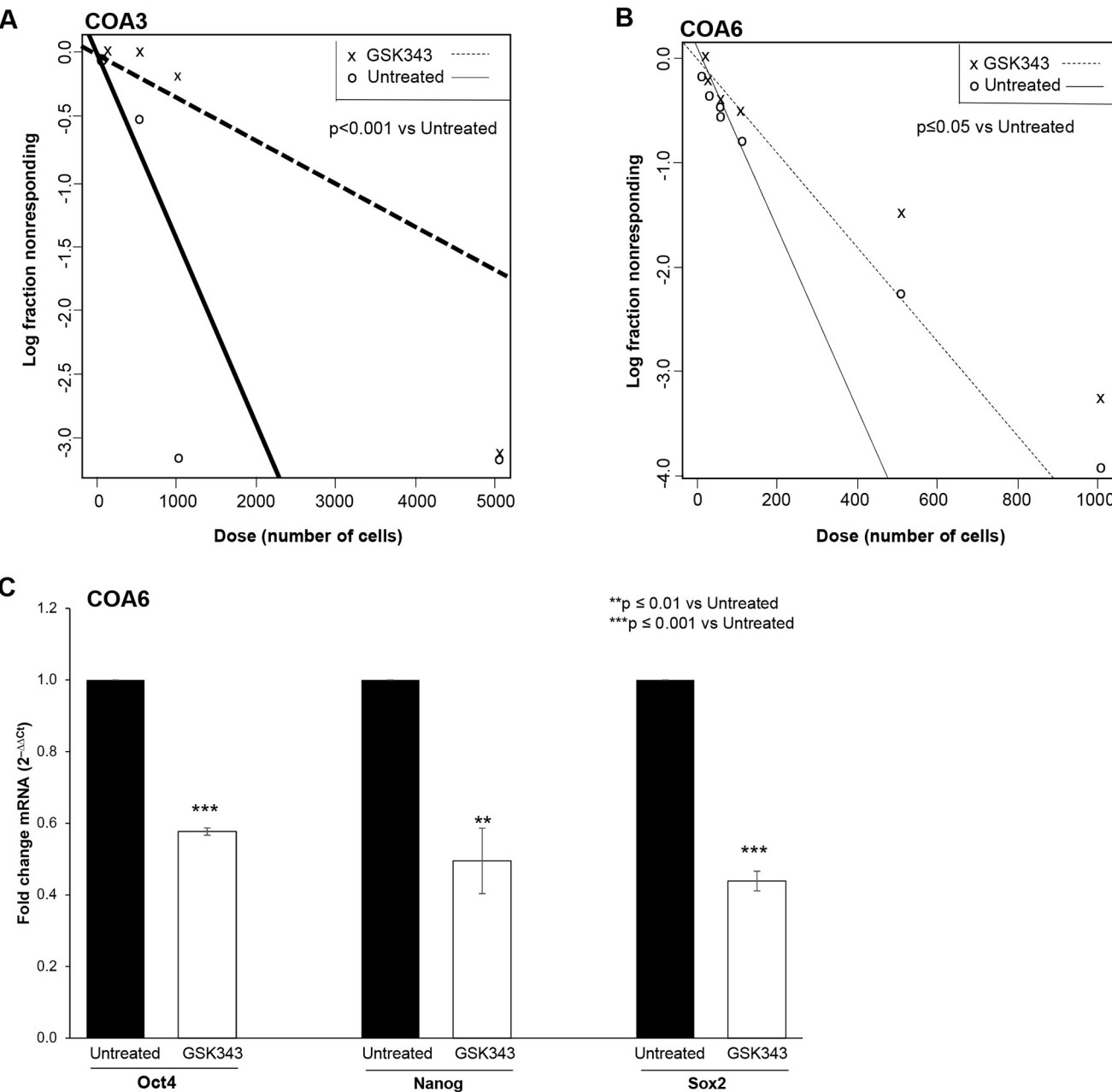

**Fig 5. GSK343 decreased tumor stemness.** (A) COA3 and (B) COA6 PDX cells were plated as single cell suspension in serum-free, conditioned media onto 96-well low attachment plates using serial dilutions with 5000, 1000, 100, 50, 20, 10, and 1 cell per well. Cells were treated with GSK343 (0, 5 μM). After one week, the number of wells containing tumorspheres in each row was counted by a single researcher and the results were analyzed using the extreme limiting dilution analysis software (http://bioinf.wehi.edu.au/software/elda/). Treatment with GSK343 significantly decreased tumorsphere formation by both the COA3 (A) and COA6 (B) cells, representing a decrease in stem cell-like phenotype. (C) COA6 cells were treated with GSK343 (0, 5 μM) for 72 hours. qPCR demonstrated a significant decrease in mRNA abundance of known stemness markers Oct4, Nanog, and Sox2 following GSK343 treatment compared to untreated cells. Data presented represent results from at least three biologic replicates.

and confocal microscopy was employed to investigate EZH2 and FAK interaction in neuroblastoma cell lines. Confocal microscopy demonstrated overlap between the two proteins indicating an interaction (Fig 6B and 6C). The Mander's overlap coefficient [17] for SK-N-AS cells was 0.66, for SK-N-BE(2) cells 0.65 (Fig 6C). These values were greater than 0, indicating

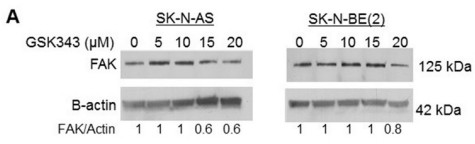

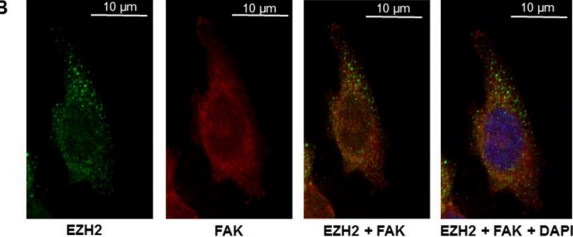

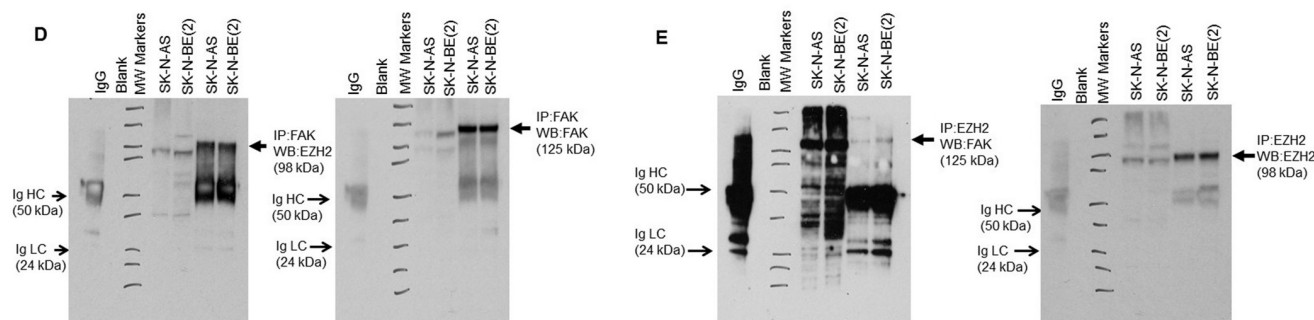

**Fig 6. EZH2 and FAK association in neuroblastoma.** (A) SK-N-AS and SK-N-BE(2) cells were treated with increasing doses of GSK343 for 24 hours. Whole cell lysates were examined with immunoblotting. There was a dose dependent decrease in FAK expression. (B) Immunofluorescence staining followed by confocal microscopy was utilized to investigate EZH2 and FAK interaction in SK-N-AS and SK-N-BE(2) cell lines. Representative photographs of SK-N-AS cells are presented. There was an overlap in staining between EZH2 (green, *first panel*) and FAK (red, *second panel*), depicted as yellow color (*third panel*), indicating an interaction. DAPI staining was completed to identify the nucleus (*fourth panel*). White scale bars represent 10 μm. (C) To quantitate overlap, Manders overlap coefficients were calculated for SK-N-AS (0.66) and SK-N-BE(2) (0.65) cells. These values were greater than 0, providing evidence of an interaction between EZH2 and FAK. (D) Immunoprecipitation with FAK followed by Western blotting for EZH2 was used to demonstrate an interaction between FAK and EZH2 (*lanes 6, 7, closed arrows*). (E) The reverse, immunoprecipitation for EZH2 followed by Western blotting for FAK revealed an interaction between EZH2 and FAK (*lanes 6, 7, closed arrows*). IgG served as a control. IgG heavy chain (HC) and light chain (LC) bands are indicated on the blots (*open arrows*). Whole cell lysates without IP were included on each of the blots (*lanes 4, 5*).

staining overlap. Co-immunoprecipitation with EZH2 and FAK antibodies was used to further demonstrate interaction between EZH2 and FAK. Immunoprecipitation with EZH2 followed by immunoblotting for FAK and the reverse, immunoprecipitation with FAK and immunoblotting for EZH2, revealed an interaction between the two proteins (Fig 6D and 6E). These results indicated that EZH2 inhibition led to decreased FAK expression, and EZH2 and FAK interact.

## Discussion

EZH2, a subunit of PCR2, utilizes SAM as a methyl group donor, leading to trimethylation of H3K27 and subsequent transcriptional silencing of target genes. Investigators have found EZH2 to be mutated and/or overexpressed in several malignancies, including prostate, lung, breast, and hepatocellular carcinoma [6], and its overexpression has been shown to be associated with a worse prognosis in neuroblastoma [7]. GSK343 is a competitive inhibitor of SAM. Verma and colleagues found that GSK343 was highly selective for EZH2 over most other methyltransferases tested, with selectivity greater than 1000-fold, with the exception of the highly homologous EZH1, where GSK343 was 60-fold more selective [26]. As such, GSK343

would not be expected to affect the expression of EZH2. In the current study, we found GSK343 treatment in neuroblastoma led to an expected decrease in the EZH2 target, tri-methylation of Histone 3 at Lysine 27 (H3K27me3), but also a decrease in EZH2 expression. Other investigators have documented this finding in various cancer cell lines including cervical cancer [27], colorectal cancer [28], gliomas [4,29], triple negative breast cancer [30], and osteosarcoma [31]. Investigators have hypothesized that changes in c-Myc may be responsible for the effects seen on EZH. C-Myc has been shown to bind to the EZH2 promoter and regulate EZH2 expression [32]. Xiong and colleagues found that GSK343 decreased expression of c-Myc in osteosarcoma cells [31], leading them to postulate that the effects of GSK343 on c-Myc were responsible for the decreased expression of EZH2 seen with GSK343 treatment [31]. We found MYCN, an important regulator of neuroblastoma, was decreased with increasing concentrations of GSK343 in SH-EP cells (supplemental information, S3). Corvetta et al demonstrated MYCN binding to EZH2 [33], so perhaps a similar mechanism may be in play in the current study and would be an avenue for further research.

The initial phenotypic changes that we explored showed that GSK343 significantly affected proliferation of neuroblastoma cells. Other investigators likewise noted these findings. EZH2 inhibitors were effective at decreasing proliferation of ovarian [34] and cervical cancer cells [27] and of glioma cells [4]. We found that GSK343 impaired neuroblastoma proliferation more than viability in the neuroblastoma cell lines. Other published studies have demonstrated similar findings. Yu and colleagues noted a significant decrease in glioma cell motility after treatment with 5 μM GSK343, but cell viability was not affected until much higher concentrations were used [4]. Similar results were seen in the study of triple negative breast cancer cells where viability never reached 50%, but proliferation was markedly diminished at 2 μM GSK343 [30]. This finding suggests that combining the anti-proliferative effects of GSK343 with a cytotoxic agent could have synergistic or additive effects. Yu et al. investigated GSK343 in combination with adriamycin, a chemotherapy agent, for triple negative breast cancer. The combination therapy resulted in decreased viability *in vitro*. They also found decreased tumor growth *in vivo* with the combination treatment when compared to either single agent alone [30]. Stazi and colleagues investigated EZH2 inhibition in combination with standard chemotherapy in glioblastoma with two EZH2 inhibitors and temozolomide. There was decreased viability with the combination treatment compared to EZH2 inhibition alone, suggesting a benefit to adding EZH2 inhibition to cytotoxic chemotherapy [35]. Combining EZH2 inhibition with cytotoxic agents in neuroblastoma could provide an exciting avenue for future studies.

In addition to decreasing cell proliferation and viability, EZH2 affected cell motility in other malignancies [5,28]. Anwar et al. demonstrated that breast cancer metastasis depended upon the phosphorylation of EZH2 at T367. Not only was phosphorylated EZH2 increased in metastatic breast cancer specimens, but they found phosphorylated EZH2 enhanced cell motility and invasion, whereas EZH2 that was not phosphorylated decreased cell proliferation [36]. EZH2 also regulated gain-of-function (GOF) mutations of p53 (mtp53) that promote cancer metastasis. In prostate cancer, the knockdown of EZH2 resulted in a decrease of the protein levels of cancer specific GOF mutations (R273H and R248W) but also decreased cell invasion [37]. Similarly, in metastatic pancreatic cancer, EZH2 increased R248W expression as well as cellular invasion. In our study, the treatment of neuroblastoma cells with GSK343 at concentrations well below the $LD_{50}$ for the compound significantly decreased neuroblastoma migration and invasion, suggesting that EZH2 may play a role in neuroblastoma metastasis that is independent from simple decreased cell viability.

It is important to note that the SK-N-AS cell line is derived from a *MYCN* non-amplified tumor while SK-N-BE(2), COA3 and COA6 are *MYCN* amplified. Further, SH-EP and WAC

(2) cell lines are isogenic for *MYCN* [12] with SH-EP being non-amplified and WAC(2) having MYCN overexpression. *MYCN* is a proto-oncogene that is associated with high-risk neuroblastoma and, when amplified, is the most important negative prognostic factor in neuroblastoma [10]. Chen et al. showed that *MYCN* amplified neuroblastoma cells expressed increased levels of EZH2 when compared to non-amplified neuroblastoma cells [8] and showed that treatment with two EZH2 inhibitors, GSK126 and JQEZ5, as well as shRNA knockdown of EZH2 led to decreased tumor growth in *MYCN* amplified long-term passage cell lines, including SK-N-BE(2) [8]. This group also found SK-N-AS, to be sensitive to EZH2 inhibition and proposed a concurrent oncogenic mutation, NRAS, as potentially driving these effects [8]. Our findings demonstrated EZH2 inhibition with GSK343 decreased the malignant phenotype in *MYCN* amplified and non-amplified cell lines. The finding that EZH2 inhibition affected both amplified and non-amplified cell lines similarly makes the continued investigation of EZH2 as a therapeutic target for high-risk neuroblastoma crucial and exciting, since the majority of children with high-risk disease does not have *MYCN* amplified tumors.

In the original studies looking at pharmacokinetics of GSK343 in rodents, it was felt that GSK343 displayed too high a clearance and might not be suitable for *in vivo* studies [26]. However, our study and those of other investigators have shown *in vivo* efficacy. In a flank murine model of cervical cancer (SiHa cells), GSK343 treatment resulted in significant reduction in tumor growth [27]. Other investigators have used GSK343 inhibition in studies of neural tumors. Stazi and colleagues utilized GSK343 and showed decreased glioblastoma tumor growth *in vivo* [35]. Yu et al demonstrated a significant increase in animal survival in mice bearing intracranial U87 glioma tumors when treated with GSK343 compared to vehicle treated animals [4]. In the current study, we found that animals treated with GSK343 had significantly decreased tumor growth, supporting the role for EZH2 inhibition in neuroblastoma. Importantly, we did not see changes in animal weights with GSK343 treatment compared to those treated with vehicle, indicating that decreased animal growth was not the etiology of decreased tumor size.

Cancer cell stemness is a crucial area of investigation due to the thought that these cells may be responsible for cancer cell self-renewal, and thereby recurrent and refractory disease [38,39]. EZH2 may promote stemness by increasing the trimethylation of H3K27 of stemness regulators, thereby decreasing these regulators at the transcriptional level. Zhang found that EZH2 repressed the stemness regulator *ATOH8* in hepatocellular carcinoma [40]. In addition, EZH2 interacted with NF-κB, promoting glioma cell proliferation, and the loss of this interaction repressed the self-renewal ability of the glioma cells [41]. Based on these findings, we hypothesized that treatment with GSK343 would lead to reduced neuroblastoma stemness. We showed the effects of GSK343 on stemness by observing decreased tumorsphere formation and decreased mRNA abundance of known stemness markers.

Utilizing confocal microscopy and immunoprecipitation, we demonstrated an interaction between FAK and EZH2. The concept of FAK binding to other proteins is not unfounded. Other investigators have shown interactions between FAK and growth factors, tumor suppressors, and other proteins affecting cancer growth. We have previously shown a direct association between FAK and VEGFR3 [42] and FAK and p53 in neuroblastoma [43]. Sieg showed that FAK associated with PDGF and EGF receptors to promote cancer cell motility [44], and Liu and others demonstrated an association of FAK with IGF-1R in pancreatic cancer cell lines leading to increased cell survival [45]. One of the first investigations of FAK and EZH2's interaction found both proteins overexpressed in endometrial carcinoma, suggesting that these proteins play a role in a more aggressive tumor phenotype and worse prognosis [22]. Gnani demonstrated an association between FAK and EZH2 by showing that FAK knockdown downregulated EZH2 and the overexpression of FAK increased EZH2 in hepatocellular

carcinoma [46]. In light of these previous studies, it would follow that FAK may interact with EZH2 to affect the neuroblastoma cell phenotype.

There are some discrepancies regarding the role of FAK in neuroblastoma. Examination of the Kocak (R2) database indicated that greater *PTK2* (*FAK*) gene abundance is associated with better overall survival in neuroblastoma [47]. However, the studies examining FAK protein expression indicated that higher protein expression was associated with worse disease including *MYCN* amplification and metastasis [48–50]. We have a few explanations for these discrepancies. First, gene expression does not always translate and equate with protein expression due to translational and post-translational modifications of gene products. Second, most of the protein data in the literature indicate a relation between high FAK expression and patients with high-risk disease or amplification of the MYCN oncogene, but these studies focused on MYCN amplified or high-risk disease. Since the gene expression datasets include all comers for the disease including very low, low and intermediate risk patients, there may be factors related to disease stratification that may contribute to the conflicting findings. In fact, in one of the earliest publications investigating FAK protein expression with immunohistochemistry in 70 neuroblastoma patient samples, patient overall or event free survival did not relate to FAK staining in a statistically significant manner. Positive IHC staining for FAK was, however, associated with MYCN amplified disease in high-risk patients.

## Conclusion

In the current study, we showed that EZH2 inhibition with GSK343 in neuroblastoma decreased proliferation, viability, motility, and tumor growth *in vivo*. The data presented also support an association between EZH2 and FAK in neuroblastoma. These findings provide evidence that EZH2 inhibitors and the mechanisms driving their anti-tumor effects warrant further investigation in neuroblastoma.

## Supporting information

**S1 Fig.**
(TIF)

**S1 Table.**
(TIF)

**S2 Table.**
(TIF)

**S1 Raw images.**
(PDF)

## Acknowledgments

The authors wish to thank Dr. Anita Hjemeland's laboratory for their assistance with the qPCR. The funding sources had no role in study design, analysis or interpretation of the data, the writing of the manuscript, or in the decision for publication submission.

## Author Contributions

**Conceptualization:** Laura V. Bownes, Adele P. Williams, Elizabeth A. Beierle.

**Data curation:** Jerry E. Stewart.

**Formal analysis:** Laura V. Bownes, Adele P. Williams, Raoud Marayati, Laura L. Stafman, Hooper Markert, Nikita Wadhwani, Elizabeth Mroczek-Musulman.

**Funding acquisition:** Elizabeth A. Beierle.

**Investigation:** Laura V. Bownes, Adele P. Williams, Raoud Marayati, Laura L. Stafman, Hooper Markert, Colin H. Quinn, Jerry E. Stewart.

**Methodology:** Elizabeth A. Beierle.

**Project administration:** Jerry E. Stewart.

**Resources:** Jamie M. Aye, Karina J. Yoon.

**Supervision:** Elizabeth A. Beierle.

**Visualization:** Laura V. Bownes, Adele P. Williams, Elizabeth A. Beierle.

**Writing – original draft:** Laura V. Bownes, Adele P. Williams.

**Writing – review & editing:** Laura V. Bownes, Adele P. Williams, Raoud Marayati, Colin H. Quinn, Elizabeth A. Beierle.

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
