## [Decision Letter · Decision Letter 0]

26 Aug 2020

PONE-D-20-19538

EZH2 inhibition decreases neuroblastoma proliferation and in vivo tumor growth

PLOS ONE

Dear Dr. Beierle,

Thank you for submitting your manuscript to PLOS ONE. After careful consideration, we feel that it has merit but does not fully meet PLOS ONE’s publication criteria as it currently stands. Therefore, we invite you to submit a revised version of the manuscript that addresses the points raised during the review process.

Please see below the comments for the reviewers. There are a number of very significant issues. Significantly you will need to address whether the effects on cell invasion and migration are due to effects in viability. Dead cells do not move. Also there are several questions related to the nature of the cell lines used (xenolines rather than PDX), clarification of why these specific ones were chosen and the limited number of cell lines tested was also a concern. Questions related to the inhibitor concentration and potential off-target effects are also raised and should be addressed.

We look forward to receiving your revised manuscript.

Kind regards,

Joe W. Ramos, Ph.D.

Academic Editor

PLOS ONE

Journal Requirements:

'Animal experiments were approved by the Institutional Animal Care and Use Committee (IACUC-09355) and were conducted within institutional, national, and NIH guidelines.'

(a) Please amend your current ethics statement to include the full name of the animal ethics committee that approved your specific study, including the full name of the affilitated institution.

(b) Once you have amended this/these statement(s) in the Methods section of the manuscript, please add the same text to the “Ethics Statement” field of the submission form (via “Edit Submission”).

For additional information about PLOS ONE submissions requirements for ethics oversight of animal work, please refer to http://journals.plos.org/plosone/s/submission-guidelines#loc-animal-research.

3. Please provide additional details regarding participant consent for obtaining the neuroblastoma tumor tissue. In the ethics statement in the Methods and online submission information, please ensure that you have specified (1) whether consent was informed and (2) what type you obtained (for instance, written or verbal, and if verbal, how it was documented and witnessed). If your study included minors, state whether you obtained consent from parents or guardians. If the need for consent was waived by the ethics committee, please include this information.

4. At this time, we request that you  please report additional details in your Methods section regarding animal care, as per our editorial guidelines:

(1) Please state the number of mice used in the both the PDX and cell line xenograft study  

Thank you for your attention to this request.

5. We noticed minor instances of text overlap with the following previous publication(s), which need to be addressed:

(1) https://laacs.org/wp-content/uploads/2019/01/oral-abstracts.pdf

(2) https://doaj.org/article/e301bf95fdb34b2097d8ae0defd2aedc

(2) https://www.sciencedirect.com/science/article/pii/S1936523318303711?via%3Dihub  

The text that needs to be addressed involves the (1, 2) Abstract and (3) Introduction section (first paragraph).

In your revision please ensure you cite all your sources (including your own works), and quote or rephrase any duplicated text outside the methods section. Further consideration is dependent on these concerns being addressed.

6. In your Methods section, please provide additional information about the participant recruitment method and the demographic details of your participants for the collection of neuroblastoma tumor tissue. Please ensure you have provided sufficient details to replicate the analyses such as: a) the recruitment date range (month and year), b) a description of any inclusion/exclusion criteria that were applied to participant recruitment, c) a table of relevant demographic details, d) a description of how participants were recruited, and e) descriptions of where participants were recruited and where the research took place.

7. At this time, we ask that you please provide scale bars on the microscopy images presented in Figure 4F and 6B and refer to the scale bar in the corresponding Figure legend.

8. PLOS ONE now requires that authors provide the original uncropped and unadjusted images underlying all blot or gel results reported in a submission’s figures or Supporting Information files. This policy and the journal’s other requirements for blot/gel reporting and figure preparation are described in detail at https://journals.plos.org/plosone/s/figures#loc-blot-and-gel-reporting-requirements and https://journals.plos.org/plosone/s/figures#loc-preparing-figures-from-image-files. When you submit your revised manuscript, please ensure that your figures adhere fully to these guidelines and provide the original underlying images for all blot or gel data reported in your submission. See the following link for instructions on providing the original image data: https://journals.plos.org/plosone/s/figures#loc-original-images-for-blots-and-gels.

9. Please amend either the abstract on the online submission form (via Edit Submission) or the abstract in the manuscript so that they are identical.

Reviewers' comments:

Reviewer's Responses to Questions

**Comments to the Author**

1. Is the manuscript technically sound, and do the data support the conclusions?

Reviewer #1: Partly

Reviewer #2: Partly

Reviewer #3: Yes

2. Has the statistical analysis been performed appropriately and rigorously? 

Reviewer #1: Yes

Reviewer #2: Yes

Reviewer #3: Yes

3. Have the authors made all data underlying the findings in their manuscript fully available?

Reviewer #1: Yes

Reviewer #2: Yes

Reviewer #3: Yes

4. Is the manuscript presented in an intelligible fashion and written in standard English?

Reviewer #1: Yes

Reviewer #2: Yes

Reviewer #3: Yes

5. Review Comments to the Author

Reviewer #1: The authors have performed a series of experiments to attempt to evaluate the efficacy of GSK343, an EZH2 inhibitor, in neuroblastoma. The manuscript presents a somewhat limited exploration of the efficacy of GSK343 and its effects on a small panel of neuroblastoma cell lines.

The manuscript is generally well written and summarizes the data well. The manuscript suffers from a few correctable weaknesses, detailed below.

1) The results compare only three neuroblastoma cell lines overall. Data from additional cell lines would provide more clarity about the relative effects and mechanisms

2) the authors should further discuss the potential for off-target effects and the relative specificity of GSK343, particularly at the tested concentrations

3) The concentrations required to demonstrate EZH2 inhibition in cell lines are quite high - are these concentrations achievable in in vivo models without toxicity? The authors should include a comment on the therapeutic window

4) the authors should elaborate on the potential mechanisms by which GSK343 reduces EZH2 expression levels

5) the doses used in the migration assay for SKNAS cells are high enough to result in cell lethality, which would lead to misinterpretation of the migration assays. Repeating assays at lower doses should be used to validate specific effects on migration. As a separate point, the dose used in SKNBE2 cells is not clear

Reviewer #2: The is a tremendous unmet clinical need to develop efficacious and safe therapies for high-risk neuroblastoma (NB). Inhibitors to EZH2 are of great interest in aggressive NB since analysis of clinical NB specimens indicate that EZH2 overexpression is associated with increased risk of relapse. This is an important area of study. Bowness et al use a small panel of NB lines to determine to what extent EZH2 inhibition as a monotherapy can block NB growth, survival, migration and invasion.

Major Comments:

The abstract is incorrectly written. The authors do not use a PDX. They use a xenoline established from the PDX. This needs to be clarified throughout the manuscript. As an example,

the title for Figure 4 legend is not correct. It should read: GSK343 decreased proliferation and survival in a xenoline derived from a human neuroblastoma PDX.

The author should include information on the NB cell lines and the xenoline regarding site of origin and relapse status.

In Figure 1, the authors state that GSK343 decreases EZH2 protein and its downstream effector – trimethylation of H3K27. There is no quantification of the EZH2 protein levels which appears to be quite modest. The antibody (C36B11) detects endogenous levels of histone H3 only when tri-methylated on Lys27. For the authors to definitively say that H3K27me levels were decreased in the presence of the GSK343, the total levels of the H3 protein need to be examined.

Fig. 2 – SK-N-AS cells were treated with 15 uM GSK343 to evaluate motility. There is no discussion if GSK343-mediated inhibition of migration and invasion could be influenced by effects on viability. In Fig. 2B, 15 uM GSK343 decreased viability. These data need to be reconciled.

Fig 2 B – In the SK-N-BE(2) cells, effects on 25 uM GSK343 on viability while evidently statistically significant, are from a biological standpoint very modest. Is this because these cells underwent cell cycle arrest? The concentration of 25 uM is very high and likely off target? Is this in the range of clinically achievable concentrations of GSK343?

In Fig 3, SK-N-BE(2) cells were used for the flank study. Why was this model chosen instead of the SK-N-AS? There is a modest decrease in tumor growth but tumors continue to grow during the dosing period. How was the dose of 10mg/kg/day chosen? There is no data presented that this dose actually decreases H3K27 trimethylation.

It appears that the mice in both groups were taken down at the same time point. The tumor weight data should be included.

Apoptosis was measured by the percentage of cells in sub-G1 phase of the cell cycle. This can include cells that were necrotic as well. Annexin V/PI staining or analysis of caspase 3/7 activation/PARP cleavage should be included to confirm these data.

In Fig 4, the COA6 xenoline was treated for 24 hours with increasing doses of GSK343 (0, 5, 10 μM) and plated in Transwell® inserts and migration monitored for 72 hrs. Treatment with GSK343 resulted in significantly decreased migration, However, 10 uM also significantly decreases viability (Fig 4B). The authors need to reconcile the possible effects of decreased viability on invasion. In the Materials and Methods, it states that the cells were allowed to invade for one week. Please discuss why this long time period was required.

The limiting dilution data in Figure 5A is not clear. Some of the untreated samples appear to associate with the “stem cell” frequency of the GSK343-treated samples.

Minor comments:

The authors state they used real-time PCR to ensure that the PDX does not have murine contamination. PDXs typically have some level of murine stromal components. Assume the authors are referring to the xenoline established from the PDX? That would not have murine contamination.

The GSK343 is characterized as a SAM-competitive inhibitor of PCR2. Are there other examples in the literature of the inhibitor destabilizing EZH2 protein levels? More information on the GSK343 and effects on EZH2 degradation should be discussed.

The authors include the following sentence twice in the MM section:

The PDX program was previously described in detail (10).

It is not clear why different concentrations of GSK343 were used in Fig 5A and 5B.

To determine if GSK343 disrupted the stem cell-like phenotype, tumorsphere forming

ability was assessed with in vitro limiting dilution assays. Conditioned COA6 media was

harvested from untreated cells in culture for the assay. Please discus why the conditioned media was used.

Fig. 6B. Please provide more details on the number of cells analyzed by the Mander’s overlap analysis.

In Fig. 6D and 6E, please denote the heavy and light chains on the westerns.

The authors discuss that the gain-of-function (GOF) mutations of p53 can promote cancer metastasis. For clarity purposed, be sure to mention that you are referring to mtp53 throughout that paragraph (pg. 23 of the discussion).

Reviewer #3: In the present study, authors used two neuroblastoma cell lines (SK-N-AS, SK-N-BE) and one PDX (COA6). The EZH2 inhibitor, GSK343, was employed and knockdown was confirmed with WB. Treatment with GSK343 led to decreased neuroblastoma cell proliferation, viability, migration, and invasion, and decreased stemness. Treatment of mice bearing SK-N-BE(2) neuroblastoma tumors with GSK343 resulted in a significant decrease in tumor growth compared to vehicle-treated animals. GSK343 was found in 2012 (ACS Med Chem Lett. 2012 Oct 19;3(12):1091-6.) and there have been several reports administered GSK343 to several tumor cell lines but not to NB cell lines according to PubMed. The findings by the authors were potentially interesting and the GSK343 effects on NB cells will be informative for NB epigenetic studies. However, several experiments need to be improved to confirm their arguments and for publication in PLOSONE.

Comments:

1. Fig.1, as a loading control, total histone H3 WB is better.

The time of incubation with GSK343 should be mentioned in the legend.

2. Fig.2e: How authors can argue that GSK343 treated cells (right panel) demonstrated significant reduction in ability to heal the scratch compared to untreated cells (left panel)?

Authors should try to quantify the reduction and to present the statistical significance.

3. Fig.3: The representative xenograft photos should be presented for readers.

4. Fig.4: Authors indicated that GSK343 significantly decreased the proliferation and viability of the NB cells. Although they argued that migration and invasion were down-regulated by GSK343, I think their experiments did not address the migration and invasion because of difference of viable cell number by GSK343 treatments.

5. Fig.6B and C: EZH2 mainly locates in nucleus and FAK (PTK2 may be better) mainly locates cytoplasm and nucleus. I can’t distinguish nucleus in Fig.6B because authors did not indicate the single DAPI staining photo. Further, the quality of IHC was not good.

6. Fig.6D and E: The quality of IP-WB experiments were not good because background signals were high. I think direct WB results by using total cell lysates were required for these experiments and MW marker lanes also should be indicated. Further, location of the Ig bands should be indicated in these IP-WBs.

7. R2 database analysis using Kocak database indicated that PTK2 low expression significantly related to the worse prognosis of NB patients. Authors need to discuss the discrepancy.

8. To study the effects of GSK343 on the FAK (PTK2) target molecules and pathways in NB cells will provide the important information for their study.

6. PLOS authors have the option to publish the peer review history of their article (what does this mean?). If published, this will include your full peer review and any attached files.

Reviewer #1: No

Reviewer #2: No

Reviewer #3: No

---

## [Author Response · Author response to Decision Letter 0]

21 Nov 2020

Date: Aug 26 2020 04:50AM

To: "Elizabeth A Beierle" elizabeth.beierle@childrensal.org

From: "PLOS ONE" plosone@plos.org

Subject: PLOS ONE Decision: Revision required [PONE-D-20-19538]

PONE-D-20-19538

EZH2 inhibition decreases neuroblastoma proliferation and in vivo tumor growth

PLOS ONE

Dear Dr. Beierle,

Thank you for submitting your manuscript to PLOS ONE. After careful consideration, we feel that it has merit but does not fully meet PLOS ONE’s publication criteria as it currently stands. Therefore, we invite you to submit a revised version of the manuscript that addresses the points raised during the review process.

Please see below the comments for the reviewers. There are a number of very significant issues. Significantly you will need to address whether the effects on cell invasion and migration are due to effects in viability. Dead cells do not move. Also there are several questions related to the nature of the cell lines used (xenolines rather than PDX), clarification of why these specific ones were chosen and the limited number of cell lines tested was also a concern. Questions related to the inhibitor concentration and potential off-target effects are also raised and should be addressed.

We look forward to receiving your revised manuscript.

Kind regards,

Joe W. Ramos, Ph.D.

Academic Editor

PLOS ONE

Journal Requirements:

'Animal experiments were approved by the Institutional Animal Care and Use Committee (IACUC-09355) and were conducted within institutional, national, and NIH guidelines.'

(a) Please amend your current ethics statement to include the full name of the animal ethics committee that approved your specific study, including the full name of the affiliated institution.

We have completed this request. 

(b) Once you have amended this/these statement(s) in the Methods section of the manuscript, please add the same text to the “Ethics Statement” field of the submission form (via “Edit Submission”).

We have completed this request in the manuscript text and the online submission.

For additional information about PLOS ONE submissions requirements for ethics oversight of animal work, please refer to http://journals.plos.org/plosone/s/submission-guidelines#loc-animal-research.

3. Please provide additional details regarding participant consent for obtaining the neuroblastoma tumor tissue. In the ethics statement in the Methods and online submission information, please ensure that you have specified (1) whether consent was informed and (2) what type you obtained (for instance, written or verbal, and if verbal, how it was documented and witnessed). If your study included minors, state whether you obtained consent from parents or guardians. If the need for consent was waived by the ethics committee, please include this information.

We have completed this request and revised the manuscript and the online submission.

4. At this time, we request that you please report additional details in your Methods section regarding animal care, as per our editorial guidelines:

(1) Please state the number of mice used in the both the PDX and cell line xenograft study 

We utilized 14 animals for the in vivo tumor growth study and 21 animals for propagating the xenolines. A statement has been added to the methods section. 

5. We noticed minor instances of text overlap with the following previous publication(s), which need to be addressed:

(1) https://laacs.org/wp-content/uploads/2019/01/oral-abstracts.pdf

(2) https://doaj.org/article/e301bf95fdb34b2097d8ae0defd2aedc

(2) https://www.sciencedirect.com/science/article/pii/S1936523318303711?via%3Dihub

The text that needs to be addressed involves the (1, 2) Abstract and (3) Introduction section (first paragraph).

These issues have been addressed in the revision. Both sections have been completely re-written.

6. In your Methods section, please provide additional information about the participant recruitment method and the demographic details of your participants for the collection of neuroblastoma tumor tissue. Please ensure you have provided sufficient details to replicate the analyses such as: a) the recruitment date range (month and year), b) a description of any inclusion/exclusion criteria that were applied to participant recruitment, c) a table of relevant demographic details, d) a description of how participants were recruited, and e) descriptions of where participants were recruited and where the research took place.

This requested information has been added to the revised Methods section and the demographics of the patients are provided in Supplemental Table 1.

7. At this time, we ask that you please provide scale bars on the microscopy images presented in Figure 4F and 6B and refer to the scale bar in the corresponding Figure legend.

Scale bars have been added for Figure 4F and 6B. 

8. PLOS ONE now requires that authors provide the original uncropped and unadjusted images underlying all blot or gel results reported in a submission’s figures or Supporting Information files. This policy and the journal’s other requirements for blot/gel reporting and figure preparation are described in detail at https://journals.plos.org/plosone/s/figures#loc-blot-and-gel-reporting-requirements and https://journals.plos.org/plosone/s/figures#loc-preparing-figures-from-image-files. When you submit your revised manuscript, please ensure that your figures adhere fully to these guidelines and provide the original underlying images for all blot or gel data reported in your submission. See the following link for instructions on providing the original image data: https://journals.plos.org/plosone/s/figures#loc-original-images-for-blots-and-gels.

The blots are available in Supporting Information, and this information has been included in the cover letter.

9. Please amend either the abstract on the online submission form (via Edit Submission) or the abstract in the manuscript so that they are identical.

We have amended the abstracts so they are identical. 

Reviewers' comments:

Reviewer's Responses to Questions

Comments to the Author

1. Is the manuscript technically sound, and do the data support the conclusions?

Reviewer #1: Partly

Reviewer #2: Partly

Reviewer #3: Yes

2. Has the statistical analysis been performed appropriately and rigorously?

Reviewer #1: Yes

Reviewer #2: Yes

Reviewer #3: Yes

3. Have the authors made all data underlying the findings in their manuscript fully available?

Reviewer #1: Yes

Reviewer #2: Yes

Reviewer #3: Yes

4. Is the manuscript presented in an intelligible fashion and written in standard English?

Reviewer #1: Yes

Reviewer #2: Yes

Reviewer #3: Yes

5. Review Comments to the Author

Reviewer #1: The authors have performed a series of experiments to attempt to evaluate the efficacy of GSK343, an EZH2 inhibitor, in neuroblastoma. The manuscript presents a somewhat limited exploration of the efficacy of GSK343 and its effects on a small panel of neuroblastoma cell lines.

The manuscript is generally well written and summarizes the data well. The manuscript suffers from a few correctable weaknesses, detailed below.

1) The results compare only three neuroblastoma cell lines overall. Data from additional cell lines would provide more clarity about the relative effects and mechanisms

We have performed additional experiments and have included data from two additional long-term passage cell lines and an additional patient-derived xenoline (PDX) adding significantly to the original submission.

2) the authors should further discuss the potential for off-target effects and the relative specificity of GSK343, particularly at the tested concentrations.

GSK343 has been shown to be highly selective for EZH2 [1]. There are reports of downstream targets such as kinases that may lead to phenotypic changes. We have added a discussion of the potential for off-target effects to the discussion section of the paper.

3) The concentrations required to demonstrate EZH2 inhibition in cell lines are quite high - are these concentrations achievable in in vivo models without toxicity? The authors should include a comment on the therapeutic window.

Although the concentrations required to affect viability are high, the concentrations that show an effect on motility and stemness are much lower. Additionally, we saw a significant effect in an in vivo murine model (Fig. 3). The concentrations utilized for those experiments were comparable to those utilized by other investigators in other cancer cell types [2, 3]. We have added a section to the discussion addressing this issue.

4) the authors should elaborate on the potential mechanisms by which GSK343 reduces EZH2 expression levels

EZH2 is a histone lysine methyltransferase that utilizes S-(S′-adenosyl)-L-methionine (SAM) as a methyl group donor, leading to trimethylation of H3K27me3 and subsequent transcriptional silencing of target genes. GSK343 is a competitive inhibitor of SAM. Verma and colleagues found that GSK343 was highly selective for EZH2 over most other methyltransferases tested, with selectivity greater than 1000-fold, with the exception of the highly homologous EZH1 where it was 60-fold more selective [1]. As such, it would not be expected that GSK343 would affect the expression of EZH2. However, in the current study, we found GSK343 treatment led to a decrease in EZH2 expression in neuroblastoma. Other investigators have also documented this finding in various cancer cell lines including cervical cancer [4], colorectal cancer [5], gliomas [2, 6], triple negative breast cancer [7], and osteosarcoma [8]. Xiong and colleagues found that GSK343 decreased expression of c-Myc in osteosarcoma cells [8]. C-Myc has been shown to bind to the EZH2 promoter and regulate EZH2 expression [9]. It was postulated that the effects of GSK343 on c-Myc were responsible for the decreased expression of EZH2 seen with GSK343 treatment [8]. Since MYCN, a member of the Myc family, is an important regulator of neuroblastoma, perhaps a similar mechanism may be in play in the current study. In Supplemental Data Figure 1, we provide an immunoblot showing that MYCN is decreased with increasing concentrations of GSK343. Clearly this mechanism will be the subject of our future investigations. The above discussion of potential mechanisms responsible for the effects of GSK343 on EZH2 expression have been added to the discussion. 

5) the doses used in the migration assay for SKNAS cells are high enough to result in cell lethality, which would lead to misinterpretation of the migration assays. Repeating assays at lower doses should be used to validate specific effects on migration. As a separate point, the dose used in SKNBE2 cells is not clear

The doses utilized for the motility experiments were chosen to be well below the calculated LD50 of GSK343 to avoid confusion with effects on motility that may be attributed merely to cell death. We have added a supplemental table (Supplemental Table 2) with the listings of the LD50 for GSK343 for each of the 4 long-term passage cell lines and the two PDXs. We have also clarified the doses utilized for each study in the manuscript and the figure legends. 

Reviewer #2: The is a tremendous unmet clinical need to develop efficacious and safe therapies for high-risk neuroblastoma (NB). Inhibitors to EZH2 are of great interest in aggressive NB since analysis of clinical NB specimens indicate that EZH2 overexpression is associated with increased risk of relapse. This is an important area of study. Bownes et al use a small panel of NB lines to determine to what extent EZH2 inhibition as a monotherapy can block NB growth, survival, migration and invasion.

Major Comments:

The abstract is incorrectly written. The authors do not use a PDX. They use a xenoline established from the PDX. This needs to be clarified throughout the manuscript. As an example,

the title for Figure 4 legend is not correct. It should read: GSK343 decreased proliferation and survival in a xenoline derived from a human neuroblastoma PDX.

Thank you for this recommendation. We have made the requested changes to the abstract and manuscript. 

The author should include information on the NB cell lines and the xenoline regarding site of origin and relapse status.

We have provided this information in the methods section and in a supplemental table (Supplemental Table 1).

In Figure 1, the authors state that GSK343 decreases EZH2 protein and its downstream effector – trimethylation of H3K27. There is no quantification of the EZH2 protein levels which appears to be quite modest. The antibody (C36B11) detects endogenous levels of histone H3 only when tri-methylated on Lys27. For the authors to definitively say that H3K27me levels were decreased in the presence of the GSK343, the total levels of the H3 protein need to be examined.

We have investigated the levels of H3 and have added these data to Figure 1. 

Fig. 2 – SK-N-AS cells were treated with 15 uM GSK343 to evaluate motility. There is no discussion if GSK343-mediated inhibition of migration and invasion could be influenced by effects on viability. In Fig. 2B, 15 uM GSK343 decreased viability. These data need to be reconciled.

We have provided data using two additional isogenic MYCN neuroblastoma cell lines. We have also added a table listing the median lethal dose (LD50) of GSK343 in all neuroblastoma cell lines and xenolines utilized in the study (Supplemental Table 2). The important issue demonstrated is that although GSK343 may not be cytotoxic at lower concentrations, it is cytostatic affecting phenotypic properties such as proliferation, motility and stemness. The alterations in these properties are noted at doses of GSK343 that were well below the calculated LD50 for the molecule. These issues have been addressed in the discussion section of the revised manuscript.

Fig 2 B – In the SK-N-BE(2) cells, effects on 25 uM GSK343 on viability while evidently statistically significant, are from a biological standpoint very modest. Is this because these cells underwent cell cycle arrest? The concentration of 25 uM is very high and likely off target? Is this in the range of clinically achievable concentrations of GSK343?

We agree that the ability of GSK343 to kill the cells is not robust. However, the significant changes in motility and stemness indicate that this molecule affects other aspects of tumorigenicity. These findings are consistent with other studies in the published literature. Yu and colleagues noted a significant decrease in glioma cell motility after treatment with 5 µM GSK343, but cell viability was not affected until much higher concentrations were used [4]. Similar results were seen in the study of triple negative breast cancer cells where viability never reached 50%, but proliferation was markedly diminished at 2 µM GSK343 [7]. We have addressed these issues in the discussion section of the revised manuscript.

In Fig 3, SK-N-BE(2) cells were used for the flank study. Why was this model chosen instead of the SK-N-AS? There is a modest decrease in tumor growth but tumors continue to grow during the dosing period. How was the dose of 10mg/kg/day chosen? There is no data presented that this dose actually decreases H3K27 trimethylation.

We utilized SK-N-BE(2) cells as our murine model since these are the cells that we have had success in growing as subcutaneous tumors previously in our laboratory. We chose not to utilize two different cell lines in order to minimize the number of animals for the study in keeping with the 3R’s (replace, reduce, refine). Since we were limiting to one cell line, we chose the MYCN amplified cell line since those tumors have the worst clinical outcome. We chose 10 mg/kg/day dosing based upon other reports in murine models available in the literature [2, 4]. These references are also available in the manuscript. We have added a section on the model and drug dosing to the discussion.

It appears that the mice in both groups were taken down at the same time point. The tumor weight data should be included.

We deeply regret that we did not obtain tumor weights at the time of euthanasia. 

Apoptosis was measured by the percentage of cells in sub-G1 phase of the cell cycle. This can include cells that were necrotic as well. Annexin V/PI staining or analysis of caspase 3/7 activation/PARP cleavage should be included to confirm these data.

We agree with this statement and have removed it from the manuscript. 

In Fig 4, the COA6 xenoline was treated for 24 hours with increasing doses of GSK343 (0, 5, 10 μM) and plated in Transwell® inserts and migration monitored for 72 hrs. Treatment with GSK343 resulted in significantly decreased migration, However, 10 uM also significantly decreases viability (Fig 4B). The authors need to reconcile the possible effects of decreased viability on invasion. In the Materials and Methods, it states that the cells were allowed to invade for one week. Please discuss why this long time period was required.

We agree that viability of both xenolines was affected by GSK343 at 10 µM, however, the LD50 of GSK343 for these two xenolines was much higher than the 10 µM concentration chosen (see Supplemental Table 2). So, we recognize that cell death may have had a small contribution to the noted decrease in migration and invasion, we believe that the phenotypic effects seen on motility are not simply due to the presence of non-viable cells, but are secondary to the effects of the drug. Other investigators have noted similar findings and we have added a section to the discussion addressing this phenomenon as mentioned above [2, 7]. We have found through multiple experiments in our laboratory with numerous xenolines that the ability of these cells to move is highly variable between xenolines. One week was chosen as this time point was the one found to provide the most consistent results in these particular xenolines.

The limiting dilution data in Figure 5A is not clear. Some of the untreated samples appear to associate with the “stem cell” frequency of the GSK343-treated samples.

The ELDA methods and analysis are outlined in a 2009 paper by Hu and Smyth [10]. Symbols on the graph represent the actual data points. The computer program fits the data into a model and calculates a line with a slope equal to the log-active cell fraction and calculates the p-values. Cells are plated with numbers from 1-5000 per well. The more stem-like the cell, the better it is at forming spheres and will form spheres at much lower number of cells per well (located in upper left side of graph). Once both treatment groups are plated with a high enough cell number (far right side of X-axis), it is not surprising that all of the wells with high number of cells will have spheres, explaining why some of the data points appear to overlap on the graphs, while the log-active cell fraction is different. 

Minor comments:

The authors state they used real-time PCR to ensure that the PDX does not have murine contamination. PDXs typically have some level of murine stromal components. Assume the authors are referring to the xenoline established from the PDX? That would not have murine contamination.

Thank you for this clarification. We utilized real-time PCR as an adjunct to short tandem repeat verification to ensure that the xenolines have not gone through a transformation to a murine tumor, which has been reported in the literature [11]. We have clarified this statement in the manuscript.

The GSK343 is characterized as a SAM-competitive inhibitor of PCR2. Are there other examples in the literature of the inhibitor destabilizing EZH2 protein levels? More information on the GSK343 and effects on EZH2 degradation should be discussed.

Thank you for this point as it is an interesting phenomenon. EZH2 is a histone lysine methyltransferase that utilizes S-(S′-adenosyl)-L-methionine (SAM) as a methyl group donor, leading to trimethylation of H3K27me3 and subsequent transcriptional silencing of target genes. GSK343 is a competitive inhibitor of SAM. Verma and colleagues found that GSK343 was highly selective for EZH2 over most other methyltransferases tested, with selectivity greater than 1000-fold, with the exception of the highly homologous EZH1 where it was 60-fold more selective [1]. As such, it would not be expected that GSK343 would affect the expression of EZH2. However, in the current study, we found GSK343 treatment led to a decrease in EZH2 expression in neuroblastoma. Other investigators have also documented this finding in various cancer cell lines including cervical cancer [4], colorectal cancer [5], gliomas [2, 6], triple negative breast cancer [7], and osteosarcoma [ 8]. Xiong and colleagues found that GSK343 decreased expression of c-Myc in osteosarcoma cells [8]. C-Myc has been shown to bind to the EZH2 promoter and regulate EZH2 expression [9]. It was postulated that the effects of GSK343 on c-Myc were responsible for the decreased expression of EZH2 seen with GSK343 treatment [8]. Since MYCN is an important regulator of neuroblastoma, perhaps a similar mechanism may be in play in the current study. We show in supplemental data Fig. 1 that MYCN was decreased following treatment with increasing doses of GSK343 in the SH-EP cells. This mechanism will be the subject of further investigations.

The authors include the following sentence twice in the MM section: The PDX program was previously described in detail (10).

Thank you for identifying this error. It has been remedied.

It is not clear why different concentrations of GSK343 were used in Fig 5A and 5B.

Thank you for recognizing this error. It has been remedied.

To determine if GSK343 disrupted the stem cell-like phenotype, tumorsphere forming

ability was assessed with in vitro limiting dilution assays. Conditioned COA6 media was

harvested from untreated cells in culture for the assay. Please discus why the conditioned media was used.

Conditioned media must be utilized when attempting to grow these xenolines cells as tumorspheres. It is hypothesized that there are growth factors and cytokines that are necessary for the cells to survive in the culture conditions necessary to evaluate tumorsphere growth.

Fig. 6B. Please provide more details on the number of cells analyzed by the Mander’s overlap analysis.

The Manders overlap analysis was completed by confocal microscopy. The computer obtained 20 images each of 15 cells and calculated the Manders overlap coefficient. The values reported in Figure 6C are the mean and SD of the biologic replicates. 

In Fig. 6D and 6E, please denote the heavy and light chains on the westerns.

Labels on Fig 6D and 6E have been added.

The authors discuss that the gain-of-function (GOF) mutations of p53 can promote cancer metastasis. For clarity purposed, be sure to mention that you are referring to mtp53 throughout that paragraph (pg. 23 of the discussion).

Thank you for this clarification. It has been corrected in the revised manuscript.

Reviewer #3: In the present study, authors used two neuroblastoma cell lines (SK-N-AS, SK-N-BE) and one PDX (COA6). The EZH2 inhibitor, GSK343, was employed and knockdown was confirmed with WB. Treatment with GSK343 led to decreased neuroblastoma cell proliferation, viability, migration, and invasion, and decreased stemness. Treatment of mice bearing SK-N-BE(2) neuroblastoma tumors with GSK343 resulted in a significant decrease in tumor growth compared to vehicle-treated animals. GSK343 was found in 2012 (ACS Med Chem Lett. 2012 Oct 19;3(12):1091-6.) and there have been several reports administered GSK343 to several tumor cell lines but not to NB cell lines according to PubMed. The findings by the authors were potentially interesting and the GSK343 effects on NB cells will be informative for NB epigenetic studies. However, several experiments need to be improved to confirm their arguments and for publication in PLOSONE.

Comments:

1. Fig.1, as a loading control, total histone H3 WB is better.

Thank you. H3 has been added to the immunoblots and to the methods section.

The time of incubation with GSK343 should be mentioned in the legend.

The time of incubation for Fig. 1 was 24 hours has been added to the results section and the legend.

2. Fig.2e: How authors can argue that GSK343 treated cells (right panel) demonstrated significant reduction in ability to heal the scratch compared to untreated cells (left panel)?

Authors should try to quantify the reduction and to present the statistical significance.

The photomicrograph in Fig. 2 G was presented as a representative example of the scratch assay plates. The scratch assays were completed in triplicate with at least three biologic replicates. The area of the scratch that was open was quantified with a computer program and reported as fold change comparing areal open after 24 hours to the area open at 0 hours, immediately following the wounding of the plate. 

3. Fig.3: The representative xenograft photos should be presented for readers.

We deeply regret that we do not have any representative xenograft photos. They were not obtained at the time of animal euthanasia. 

4. Fig.4: Authors indicated that GSK343 significantly decreased the proliferation and viability of the NB cells. Although they argued that migration and invasion were down-regulated by GSK343, I think their experiments did not address the migration and invasion because of difference of viable cell number by GSK343 treatments.

In Figure 2 and Figure 4, the concentrations of GSK343 chosen for the motility studies was well below the calculated LD50 of the drug (Supplemental Table 2). Although GSK343 had some effect on viability at lower concentrations, we believe that most of the effect on the motility phenotype was not attributed to decreased cell death, but from the effects of the drug, since these effects were noted at lower concentrations. The same was seen for proliferation. Proliferation was significantly decreased at concentrations of GSK343 that were below the LD50 concentrations. 

5. Fig.6B and C: EZH2 mainly locates in nucleus and FAK (PTK2 may be better) mainly locates cytoplasm and nucleus. I can’t distinguish nucleus in Fig.6B because authors did not indicate the single DAPI staining photo. Further, the quality of IHC was not good.

We have provided new IHC pictures with DAPI added for Figure 6 B and the figure legend has been revised. The nucleus is represented by the blue DAPI staining. 

6. Fig.6D and E: The quality of IP-WB experiments were not good because background signals were high. I think direct WB results by using total cell lysates were required for these experiments and MW marker lanes also should be indicated. Further, location of the Ig bands should be indicated in these IP-WBs.

We agree that the background signal was high on the IP Westerns. Unfortunately, since these studies required an inordinate amount of protein (500 µg) to be loaded, it led to significant background. However, we believe that these blots are important for the study since they demonstrate an interaction between the two proteins, FAK and EZH2 that has not been demonstrated previously in neuroblastoma. These blots also lend credence to the concept that alterations in FAK are responsible for some of the phenotypic changes seen following treatment with GSK343. 

7. R2 database analysis using Kocak database indicated that PTK2 low expression significantly related to the worse prognosis of NB patients. Authors need to discuss the discrepancy.

We have found in our laboratory that increased FAK activation and expression was associated with worse prognosis and with amplification of MYCN in neuroblastoma [12, 13]. 

8. To study the effects of GSK343 on the FAK (PTK2) target molecules and pathways in NB cells will provide the important information for their study.

We agree that these will be important areas to pursue in future studies and are the subject of our ongoing studies. 

References for reviewers’ queries.

1. Verma SK, Tian X, LaFrance LV, Duquenne C, ´ Suarez DP, Newlander KA, et al. Identification of potent, selective, cell-active inhibitors of the histone lysine methyltransferase EZH2. ACS Med. Chem. Lett. 2012;3:1091-1096.

2. Yu T, Wang Y, Hu Q, Wu W, Wu Y, Wei W, et al. The EZH2 inhibitor GSK343 suppresses cancer stem-like phenotypes and reverses mesenchymal transition in glioma cells. Oncotarget. 2017;8(58):98348-59. Epub 2017/12/13. 

3. Zhou J, Huang S, Wang Z, Huang J, Xu L, Tang X, et al. Targeting EZH2 histone methyltransferase activity alleviates experimental intestinal inflammation. Nat Commun. 2019;10(1):2427. Epub 2019/06/05.

4. Ding M, Zhang H, Li Z, Wang C, Chen J, Shi L, et al. The polycomb group protein enhancer of zeste 2 is a novel therapeutic target for cervical cancer. Clin Exp Pharmacol Physiol. 2015;42:458–464. 

5. Ying L, Yan F, Williams BR, Xu P, Li X, Zhao Y, et al. Epigallocatechin-3-gallate and EZH2 inhibitor GSK343 have similar inhibitory effects and mechanisms of action on colorectal cancer cells. Clin Exp Pharmacol Physiol. 2018;45(1):58-67. Epub 2017 Nov 2. 

6. Mohammad F, Weissmann S, Leblanc B, Pandey DP, Højfeldt JW, Comet I, et al. EZH2 is a potential therapeutic target for H3K27M-mutant pediatric gliomas. Nat Med. 2017;23(4):483-492. Epub 2017 Feb 27. 

7. Yu Y, Qi J, Xiong J, Jiang L, Cui D, He J, et al. Epigenetic co-deregulation of EZH2/TET1 is a senescence-countering, actionable vulnerability in triple-negative breast cancer. Theranostics. 2019;9(3):761-777. eCollection 2019.

8. Xiong X, Zhang J, Liang W, Cao W, Qin S, Dai L, et al. Fuse-binding protein 1 is a target of the EZH2 inhibitor GSK343, in osteosarcoma cells. Int J Oncol. 2016;49(2):623-628. Epub 2016 May 27.

9. Kaur M, Cole MD. MYC acts via the PTEN tumor suppressor to elicit autoregulation and genome-wide gene repression by activation of the Ezh2 methyltransferase. Cancer Res. 2013;73:695–705. 

10. Hu Y, Smyth GK. ELDA: Extreme limiting dilution analysis for comparing depleted and enriched populations in stem cell and other assays. J Immunol Methods. 2009;347(1-2):70-8.

11. Ring E, Li R, Moore BP, et al. Newly characterized murine undifferentiated sarcoma models sensitive to virotherapy with oncolytic HSV-1 M002. Mol Ther Oncolytics. 2017;7:27-36.

12. Beierle EA, Trujillo A, Nagaram A, et al. N-MYC regulates focal adhesion kinase expression in human neuroblastoma. J Biol Chem. 2007;282(17):12503-16.

13. Beierle EA, Massoll NA, Hartwich J, et al. Focal adhesion kinase expression in human neuroblastoma: immunohistochemical and real-time PCR analyses. Clin Cancer Res. 2008;14(11):3299-305.

---

## [Decision Letter · Decision Letter 1]

9 Dec 2020

PONE-D-20-19538R1

EZH2 inhibition decreases neuroblastoma proliferation and in vivo tumor growth

PLOS ONE

Dear Dr. Beierle,

Thank you for submitting your manuscript to PLOS ONE. After careful consideration, we feel that it has merit but does not fully meet PLOS ONE’s publication criteria as it currently stands. Therefore, we invite you to submit a revised version of the manuscript that addresses the points raised during the review process.

The revised manuscript is much improved. There remain a few issues noted below that I feel are significant enough to require attention. Specifically reviewer #2 points out an issue with the conclusion of Figure 1 and Reviewer #2 notes that the data supporting interaction of EZH2 and FAK remains poor in Figure 6D/E and some discussion is required for Figure 7. I also would expect to see the lysates run on the same gels and ideally a much improved signal to noise ratio. I believe these are generally straightforward requests and will better support the conclusions. 

We look forward to receiving your revised manuscript.

Kind regards,

Joe W. Ramos, Ph.D.

Academic Editor

PLOS ONE

Reviewers' comments:

Reviewer's Responses to Questions

**Comments to the Author**

1. If the authors have adequately addressed your comments raised in a previous round of review and you feel that this manuscript is now acceptable for publication, you may indicate that here to bypass the “Comments to the Author” section, enter your conflict of interest statement in the “Confidential to Editor” section, and submit your "Accept" recommendation.

Reviewer #1: All comments have been addressed

Reviewer #2: (No Response)

Reviewer #3: (No Response)

2. Is the manuscript technically sound, and do the data support the conclusions?

Reviewer #1: Yes

Reviewer #2: Yes

Reviewer #3: Partly

3. Has the statistical analysis been performed appropriately and rigorously? 

Reviewer #1: Yes

Reviewer #2: Yes

Reviewer #3: Yes

4. Have the authors made all data underlying the findings in their manuscript fully available?

Reviewer #1: Yes

Reviewer #2: Yes

Reviewer #3: Yes

5. Is the manuscript presented in an intelligible fashion and written in standard English?

Reviewer #1: Yes

Reviewer #2: Yes

Reviewer #3: Yes

6. Review Comments to the Author

Reviewer #1: All of my comments and questions have been addressed

Reviewer #2: Overall, the authors have carefully addressed the reviewers’ comments and the manuscript is ready for publication after one minor revision of the text. The interpretation of data in Figure 1 needs clarification (lines 318-327). The title of Figure 1 is “EZH2 inhibitor, GSK343, decreased expression of H3K27me3.” This is not correct for the authors now show that H3 expression was not changed in the presence of GSK343. Based on inhibitor mechanism, the EZH2 inhibitor decreased the methylation of Histone 3 at Lysine 27. This should be corrected in the title for Figure 1, the legend title for Figure 1, results section for Figure 1 as well as in the discussion text for Figure 1 (lines 497- 499).

New title suggestion: EZH2 inhibitor, GSK343, decreased tri-methylation of Histone 3 at Lysine 27.

Reviewer #3: In the revised version, authors modified several parts of the paper appropriately, however, my comments 6 and 7 are still unresolved.

1. Fig.1, as a loading control, total histone H3 WB is better.

＞Thank you. H3 has been added to the immunoblots and to the methods section.

The time of incubation with GSK343 should be mentioned in the legend.

＞The time of incubation for Fig. 1 was 24 hours has been added to the results section and the legend.

I accept their answer.

2. Fig.2e: How authors can argue that GSK343 treated cells (right panel) demonstrated significant reduction in ability to heal the scratch compared to untreated cells (left panel)?

Authors should try to quantify the reduction and to present the statistical significance.

>The photomicrograph in Fig. 2 G was presented as a representative example of the scratch assay plates. The scratch assays were completed in triplicate with at least three biologic replicates. The area of the scratch that was open was quantified with a computer program and reported as fold change comparing areal open after 24 hours to the area open at 0 hours, immediately following the wounding of the plate.

I accept their answer.

3. Fig.3: The representative xenograft photos should be presented for readers.

We deeply regret that we do not have any representative xenograft photos. They were not obtained at the time of animal euthanasia.

I accept their answer.

4. Fig.4: Authors indicated that GSK343 significantly decreased the proliferation and viability of the NB cells.

Although they argued that migration and invasion were down-regulated by GSK343, I think their experiments did not address the migration and invasion because of difference of viable cell number by GSK343 treatments.

>In Figure 2 and Figure 4, the concentrations of GSK343 chosen for the motility studies was well below the calculated LD50 of the drug (Supplemental Table 2). Although GSK343 had some effect on viability at lower concentrations, we believe that most of the effect on the motility phenotype was not attributed to decreased cell death, but from the effects of the drug, since these effects were noted at lower concentrations. The same was seen for proliferation. Proliferation was significantly decreased at concentrations of GSK343 that were below the LD50 concentrations.

I accept their answer.

5. Fig.6B and C: EZH2 mainly locates in nucleus and FAK (PTK2 may be better) mainly locates cytoplasm and nucleus. I can’t distinguish nucleus in Fig.6B because authors did not indicate the single DAPI staining photo. Further, the quality of IHC was not good.

>We have provided new IHC pictures with DAPI added for Figure 6 B and the figure legend has been revised. The nucleus is represented by the blue DAPI staining.

I accept their modification.

6. Fig.6D and E: The quality of IP-WB experiments were not good because background signals were high. I think direct WB results by using total cell lysates were required for these experiments and MW marker lanes also should be indicated. Further, location of the Ig bands should be indicated in these IP-WBs.

>We agree that the background signal was high on the IP Westerns. Unfortunately, since these studies required an inordinate amount of protein (500 µg) to be loaded, it led to significant background. However, we believe that these blots are important for the study since they demonstrate an interaction between the two proteins, FAK and EZH2 that has not been demonstrated previously in neuroblastoma. These blots also lend credence to the concept that alterations in FAK are responsible for some of the phenotypic changes seen following treatment with GSK343.

I agree that interaction between EZH2 and FAK is important for their work. However, the present IP-WB results has too low quality to confirm that. I do require direct WB results by using total cell lysates in the same gel experiments and reduction of the background. Further, IgG lane should not be cut form the sample lanes. My suggestion is anti-EZH2 ab from Millipore may be better than the ab they used for WB. If they can not improve their endogenous IP-WBs, they need to think about detection of the interaction by transfection of one of the molecules in NB cells.

7. R2 database analysis using Kocak database indicated that PTK2 low expression significantly related to the worse prognosis of NB patients. Authors need to discuss the discrepancy.

>We have found in our laboratory that increased FAK activation and expression was associated with worse prognosis and with amplification of MYCN in neuroblastoma [12, 13].

Ref 12 is the following: Augmented MYCN Expression Advances the Malignant Phenotype of Human Neuroblastoma Cells: Evidence for Induction of Autocrine Growth Factor Activity? I could not find the K-M analysis in that. Further, not only KOCAK DB (n=649) but also SEQC DB (n=498) indicated that PTK2 low expression significantly related to the worse prognosis of NB patients. Authors need to discuss the discrepancy in the revised version.

8. To study the effects of GSK343 on the FAK (PTK2) target molecules and pathways in NB cells will provide the important information for their study.

>We agree that these will be important areas to pursue in future studies and are the subject of our ongoing studies.

I accept the argument.

7. PLOS authors have the option to publish the peer review history of their article (what does this mean?). If published, this will include your full peer review and any attached files.

Reviewer #1: No

Reviewer #2: No

Reviewer #3: No

---

## [Author Response · Author response to Decision Letter 1]

7 Jan 2021

The revised manuscript is much improved. There remain a few issues noted below that I feel are significant enough to require attention. Specifically reviewer #2 points out an issue with the conclusion of Figure 1 and Reviewer #2 notes that the data supporting interaction of EZH2 and FAK remains poor in Figure 6D/E and some discussion is required for Figure 7. I also would expect to see the lysates run on the same gels and ideally a much improved signal to noise ratio. I believe these are generally straightforward requests and will better support the conclusions. 

Thank you for your comments and suggestions. We have revised the description of Figure 1 based upon the suggestions of Reviewer #1. We have also performed additional experiments again showing an interaction between FAK and EZH2. These IP Westerns have been provided in revised Figure 6. We believe that they are of better quality. We have also provided a detailed explanation for the queries posed by Reviewer #2. 

 Reviewers' comments:

Reviewer's Responses to Questions

 Comments to the Author

1. If the authors have adequately addressed your comments raised in a previous round of review and you feel that this manuscript is now acceptable for publication, you may indicate that here to bypass the “Comments to the Author” section, enter your conflict of interest statement in the “Confidential to Editor” section, and submit your "Accept" recommendation.

Reviewer #1: All comments have been addressed

Reviewer #2: (No Response)

Reviewer #3: (No Response)

2. Is the manuscript technically sound, and do the data support the conclusions?

 Reviewer #1: Yes

Reviewer #2: Yes

Reviewer #3: Partly

See revisions

3. Has the statistical analysis been performed appropriately and rigorously? 

 Reviewer #1: Yes

Reviewer #2: Yes

Reviewer #3: Yes 

4. Have the authors made all data underlying the findings in their manuscript fully available?

Reviewer #1: Yes

Reviewer #2: Yes

Reviewer #3: Yes 

5. Is the manuscript presented in an intelligible fashion and written in standard English?

Reviewer #1: Yes

Reviewer #2: Yes

Reviewer #3: Yes 

6. Review Comments to the Author

Reviewer #1: All of my comments and questions have been addressed

Thank you.

Reviewer #2: Overall, the authors have carefully addressed the reviewers’ comments and the manuscript is ready for publication after one minor revision of the text. The interpretation of data in Figure 1 needs clarification (lines 318-327). The title of Figure 1 is “EZH2 inhibitor, GSK343, decreased expression of H3K27me3.” This is not correct for the authors now show that H3 expression was not changed in the presence of GSK343. Based on inhibitor mechanism, the EZH2 inhibitor decreased the methylation of Histone 3 at Lysine 27. This should be corrected in the title for Figure 1, the legend title for Figure 1, results section for Figure 1 as well as in the discussion text for Figure 1 (lines 497- 499).

New title suggestion: EZH2 inhibitor, GSK343, decreased tri-methylation of Histone 3 at Lysine 27.

Thank you. We made these changes to the Results section, the Legend title for Figure 1 and the text for Figure 1 legend. We have also made the changes to the discussion text lines 501-503.

Reviewer #3: In the revised version, authors modified several parts of the paper appropriately, however, my comments 6 and 7 are still unresolved.

1. Fig.1, as a loading control, total histone H3 WB is better.

＞Thank you. H3 has been added to the immunoblots and to the methods section.

The time of incubation with GSK343 should be mentioned in the legend.

＞The time of incubation for Fig. 1 was 24 hours has been added to the results section and the legend.

I accept their answer.

Thank you.

2. Fig.2e: How authors can argue that GSK343 treated cells (right panel) demonstrated significant reduction in ability to heal the scratch compared to untreated cells (left panel)?

Authors should try to quantify the reduction and to present the statistical significance.

>The photomicrograph in Fig. 2 G was presented as a representative example of the scratch assay plates. The scratch assays were completed in triplicate with at least three biologic replicates. The area of the scratch that was open was quantified with a computer program and reported as fold change comparing areal open after 24 hours to the area open at 0 hours, immediately following the wounding of the plate.

I accept their answer.

Thank you.

3. Fig.3: The representative xenograft photos should be presented for readers.

We deeply regret that we do not have any representative xenograft photos. They were not obtained at the time of animal euthanasia.

I accept their answer.

Thank you.

4. Fig.4: Authors indicated that GSK343 significantly decreased the proliferation and viability of the NB cells.

Although they argued that migration and invasion were down-regulated by GSK343, I think their experiments did not address the migration and invasion because of difference of viable cell number by GSK343 treatments.

>In Figure 2 and Figure 4, the concentrations of GSK343 chosen for the motility studies was well below the calculated LD50 of the drug (Supplemental Table 2). Although GSK343 had some effect on viability at lower concentrations, we believe that most of the effect on the motility phenotype was not attributed to decreased cell death, but from the effects of the drug, since these effects were noted at lower concentrations. The same was seen for proliferation. Proliferation was significantly decreased at concentrations of GSK343 that were below the LD50 concentrations.

I accept their answer.

Thank you.

5. Fig.6B and C: EZH2 mainly locates in nucleus and FAK (PTK2 may be better) mainly locates cytoplasm and nucleus. I can’t distinguish nucleus in Fig.6B because authors did not indicate the single DAPI staining photo. Further, the quality of IHC was not good.

>We have provided new IHC pictures with DAPI added for Figure 6 B and the figure legend has been revised. The nucleus is represented by the blue DAPI staining.

I accept their modification.

Thank you.

6. Fig.6D and E: The quality of IP-WB experiments were not good because background signals were high. I think direct WB results by using total cell lysates were required for these experiments and MW marker lanes also should be indicated. Further, location of the Ig bands should be indicated in these IP-WBs.

>We agree that the background signal was high on the IP Westerns. Unfortunately, since these studies required an inordinate amount of protein (500 µg) to be loaded, it led to significant background. However, we believe that these blots are important for the study since they demonstrate an interaction between the two proteins, FAK and EZH2 that has not been demonstrated previously in neuroblastoma. These blots also lend credence to the concept that alterations in FAK are responsible for some of the phenotypic changes seen following treatment with GSK343.

I agree that interaction between EZH2 and FAK is important for their work. However, the present IP-WB results has too low quality to confirm that. I do require direct WB results by using total cell lysates in the same gel experiments and reduction of the background. Further, IgG lane should not be cut form the sample lanes. My suggestion is anti-EZH2 ab from Millipore may be better than the ab they used for WB. If they can not improve their endogenous IP-WBs, they need to think about detection of the interaction by transfection of one of the molecules in NB cells.

Thank you for your comments and for your appreciation of the difficulties in obtaining good quality IP-WBs. We have repeated the IP Western experiments and have provided these new immunoblots in the revised manuscript in Figure 6 D, E. They again support an interaction between FAK and EZH2. We also feel that they are of excellent quality. The previous blots remain in supplemental data for reference. 

7. R2 database analysis using Kocak database indicated that PTK2 low expression significantly related to the worse prognosis of NB patients. Authors need to discuss the discrepancy.

>We have found in our laboratory that increased FAK activation and expression was associated with worse prognosis and with amplification of MYCN in neuroblastoma [12, 13].

Ref 12 is the following: Augmented MYCN Expression Advances the Malignant Phenotype of Human Neuroblastoma Cells: Evidence for Induction of Autocrine Growth Factor Activity? I could not find the K-M analysis in that. Further, not only KOCAK DB (n=649) but also SEQC DB (n=498) indicated that PTK2 low expression significantly related to the worse prognosis of NB patients. Authors need to discuss the discrepancy in the revised version.

Thank you for your thorough investigation of the Kocak (R2) and other gene databases. We agree that a discrepancy exists between the gene expression data and the data that address the protein expression and phosphorylation status of the FAK protein in neuroblastoma. The Kocak (R2) and other databases, that examined gene expression, revealed that lower FAK (PTK2) gene expression is associated with worse overall survival in neuroblastoma. However, the studies examining FAK protein expression have indicated that higher protein expression was associated with worse disease including MYCN amplification and metastasis [1-3]. We have a few explanations for these discrepancies. First, gene expression does not always translate and equate with protein expression due to translational and post-translational modifications of gene products. Second, most of the protein data in the literature indicate a relation between high FAK expression and patients with high-risk disease or amplification of the MYCN oncogene, but these studies focused on MYCN amplified or high-risk disease and often did not include patients with low or intermediate risk tumors. Since the gene expression datasets include all comers for the disease including very low, low and intermediate risk patients, there may be factors related to disease stratification that may contribute to the conflicting findings. In fact, in one of our earliest publications investigating FAK protein expression with immunohistochemistry in 70 patient samples, patient overall or event free survival did not relate to FAK staining in a statistically significant manner. Positive IHC staining for FAK was, however, associated with MYCN amplified disease in high-risk patients [3]. We have added this discussion to the discussion section lines 611-627 of the revised manuscript.

8. To study the effects of GSK343 on the FAK (PTK2) target molecules and pathways in NB cells will provide the important information for their study.

>We agree that these will be important areas to pursue in future studies and are the subject of our ongoing studies.

I accept the argument.

Thank you.

 References for revisions

1. Kratimenos P, Koutroulis I, Syriopoulou V, Michailidi C, Delivoria-Papadopoulos M, Klijanienko J, Theocharis S. FAK-Src-paxillin system expression and disease outcome in human neuroblastoma. Pediatr Hematol Oncol. 2017 May;34(4):221-230. doi: 10.1080/08880018.2017.1360969. Epub 2017 Oct 17. PMID: 29040002

2. Lee S, Qiao J, Paul P, O'Connor KL, Evers MB, Chung DH. FAK is a critical regulator of neuroblastoma liver metastasis. Oncotarget. 2012 Dec;3(12):1576-87. doi: 10.18632/oncotarget.732. PMID: 23211542

3. Beierle EA, Massoll NA, Hartwich J, Kurenova EV, Golubovskaya VM, Cance WG, McGrady P, London WB. Focal adhesion kinase expression in human neuroblastoma: immunohistochemical and real-time PCR analyses. Clin Cancer Res. 2008 Jun 1;14(11):3299-305. doi: 10.1158/1078-0432.CCR-07-1511. PMID: 18519756

---

## [Decision Letter · Decision Letter 2]

19 Jan 2021

EZH2 inhibition decreases neuroblastoma proliferation and in vivo tumor growth

PONE-D-20-19538R2

Dear Dr. Beierle,

We’re pleased to inform you that your manuscript has been judged scientifically suitable for publication and will be formally accepted for publication once it meets all outstanding technical requirements.

Kind regards,

Joe W. Ramos, Ph.D.

Academic Editor

PLOS ONE

Additional Editor Comments (optional):

The only remaining comments relates to Figure 6 D, E and it is editorial. The reviewer noted the authors should mention the detail of each lanes in the figure legend, e.g. lanes 5 and 6 are total cell lysates.

Reviewers' comments:

Reviewer's Responses to Questions

**Comments to the Author**

1. If the authors have adequately addressed your comments raised in a previous round of review and you feel that this manuscript is now acceptable for publication, you may indicate that here to bypass the “Comments to the Author” section, enter your conflict of interest statement in the “Confidential to Editor” section, and submit your "Accept" recommendation.

Reviewer #2: All comments have been addressed

Reviewer #3: All comments have been addressed

2. Is the manuscript technically sound, and do the data support the conclusions?

Reviewer #2: Yes

Reviewer #3: Yes

3. Has the statistical analysis been performed appropriately and rigorously? 

Reviewer #2: Yes

Reviewer #3: Yes

4. Have the authors made all data underlying the findings in their manuscript fully available?

Reviewer #2: Yes

Reviewer #3: Yes

5. Is the manuscript presented in an intelligible fashion and written in standard English?

Reviewer #2: Yes

Reviewer #3: Yes

6. Review Comments to the Author

Reviewer #2: The authors have carefully addressed the reviewers’ comments. The study is now acceptable for publication.

Reviewer #3: 6. Fig.6D and E: The quality of IP-WB experiments were not good because background signals were high. I think direct WB results by using total cell lysates were required for these experiments and MW marker lanes also should be indicated. Further, location of the Ig bands should be indicated in these IP-WBs. >We agree that the background signal was high on the IP Westerns. Unfortunately, since these studies required an inordinate amount of protein (500 µg) to be loaded, it led to significant background. However, we believe that these blots are important for the study since they demonstrate an interaction between the two proteins, FAK and EZH2 that has not been demonstrated previously in neuroblastoma. These blots also lend credence to the concept that alterations in FAK are responsible for some of the phenotypic changes seen following treatment with GSK343. I agree that interaction between EZH2 and FAK is important for their work. However, the present IP-WB results has too low quality to confirm that. I do require direct WB results by using total cell lysates in the same gel experiments and reduction of the background. Further, IgG lane should not be cut form the sample lanes. My suggestion is anti-EZH2 ab from Millipore may be better than the ab they used for WB. If they can not improve their endogenous IP-WBs, they need to think about detection of the interaction by transfection of one of the molecules in NB cells. Thank you for your comments and for your appreciation of the difficulties in obtaining good quality IP-WBs. We have repeated the IP Western experiments and have provided these new immunoblots in the revised manuscript in Figure 6 D, E. They again support an interaction between FAK and EZH2. We also feel that they are of excellent quality. The previous blots remain in supplemental data for reference

>> The new immunoblots in the revised manuscript in Figure 6 D, E are acceptable. However, authors should mention the detail of each lanes in the figure legend, e.g. lanes 5 and 6 are total cell lysates.

7. R2 database analysis using Kocak database indicated that PTK2 low expression significantly related to the worse prognosis of NB patients. Authors need to discuss the discrepancy. >We have found in our laboratory that increased FAK activation and expression was associated with worse prognosis and with amplification of MYCN in neuroblastoma [12, 13]. Ref 12 is the following: Augmented MYCN Expression Advances the Malignant Phenotype of Human Neuroblastoma Cells: Evidence for Induction of Autocrine Growth Factor Activity? I could not find the K-M analysis in that. Further, not only KOCAK DB (n=649) but also SEQC DB (n=498) indicated that PTK2 low expression significantly related to the worse prognosis of NB patients. Authors need to discuss the discrepancy in the revised version. Thank you for your thorough investigation of the Kocak (R2) and other gene databases. We agree that a discrepancy exists between the gene expression data and the data that address the protein expression and phosphorylation status of the FAK protein in neuroblastoma. The Kocak (R2) and other databases, that examined gene expression, revealed that lower FAK (PTK2) gene expression is associated with worse overall survival in neuroblastoma. However, the studies examining FAK protein expression have indicated that higher protein expression was associated with worse disease including MYCN amplification and metastasis [1-3]. We have a few explanations for these discrepancies. First, gene expression does not always translate and equate with protein expression due to translational and post-translational modifications of gene products. Second, most of the protein data in the literature indicate a relation between high FAK expression and patients with high-risk disease or amplification of the MYCN oncogene, but these studies focused on MYCN amplified or high-risk disease and often did not include patients with low or intermediate risk tumors. Since the gene expression datasets include all comers for the disease including very low, low and intermediate risk patients, there may be factors related to disease stratification that may contribute to the conflicting findings. In fact, in one of our earliest publications investigating FAK protein expression with immunohistochemistry in 70 patient samples, patient overall or event free survival did not relate to FAK staining in a statistically significant manner. Positive IHC staining for FAK was, however, associated with MYCN amplified disease in high-risk patients [3]. We have added this discussion to the discussion section lines 611-627 of the revised manuscript.

>> I accept the discussion.

7. PLOS authors have the option to publish the peer review history of their article (what does this mean?). If published, this will include your full peer review and any attached files.

Reviewer #2: **Yes: **Karen E. Pollok

Reviewer #3: No

---

## [Editor Report · Acceptance letter]

17 Feb 2021

PONE-D-20-19538R2 

EZH2 inhibition decreases neuroblastoma proliferation and *in vivo* tumor growth 

Dear Dr. Beierle:

I'm pleased to inform you that your manuscript has been deemed suitable for publication in PLOS ONE. Congratulations! Your manuscript is now with our production department. 

Kind regards, 

on behalf of

Dr. Joe W. Ramos 

Academic Editor

PLOS ONE